# Combining robotic training and inactivation of the healthy hemisphere restores pre-stroke motor patterns in mice

Cristina Spalletti[1], Claudia Alia[1,2], Stefano Lai[3], Alessandro Panarese[3], Sara Conti[3], Silvestro Micera[3,4†], Matteo Caleo[1†*]

[1]CNR Neuroscience Institute, Pisa, Italy; [2]Scuola Normale Superiore, Pisa, Italy; [3]Scuola Superiore Sant'Anna, Translational Neural Engineering Area, The BioRobotics Institute, Pontedera, Italy; [4]Bertarelli Foundation Chair in Translational NeuroEngineering Laboratory, Ecole Polytechnique Federale de Lausanne (EPFL), Center for Neuroprosthetics and Institute of Bioengineering, Lausanne, Switzerland

**Abstract** Focal cortical stroke often leads to persistent motor deficits, prompting the need for more effective interventions. The efficacy of rehabilitation can be increased by 'plasticity-stimulating' treatments that enhance experience-dependent modifications in spared areas. Transcallosal pathways represent a promising therapeutic target, but their role in post-stroke recovery remains controversial. Here, we demonstrate that the contralesional cortex exerts an enhanced interhemispheric inhibition over the perilesional tissue after focal cortical stroke in mouse forelimb motor cortex. Accordingly, we designed a rehabilitation protocol combining intensive, repeatable exercises on a robotic platform with reversible inactivation of the contralesional cortex. This treatment promoted recovery in general motor tests and in manual dexterity with remarkable restoration of pre-lesion movement patterns, evaluated by kinematic analysis. Recovery was accompanied by a reduction of transcallosal inhibition and 'plasticity brakes' over the perilesional tissue. Our data support the use of combinatorial clinical therapies exploiting robotic devices and modulation of interhemispheric connectivity.

DOI: https://doi.org/10.7554/eLife.28662.001

**\*For correspondence:**
caleo@in.cnr.it

[†]These authors contributed equally to this work

**Competing interests:** The authors declare that no competing interests exist.

## Introduction

Focal cortical stroke in motor cortex often leads to persistent motor deficits that strongly impact the patients' quality of life. There is obviously great interest in developing new interventions that stimulate neuroplastic processes thus enhancing functional recovery. The standard approach for stroke rehabilitation is physical therapy, which can be delivered either by a therapist or via mechatronic devices. In particular, robots represent a valid approach to increase the amount and repeatability of the exercises (*Lo et al., 2010*; *Klamroth-Marganska et al., 2014*; *Reinkensmeyer et al., 2016*), allowing for highly standardized therapeutic protocols (*Loureiro et al., 2011*), with competitive costs compared to conventional therapy (*Wagner et al., 2011*). Importantly, they collect objective and quantitative data about the performance of each patient, characterized by different kinetic and kinematic parameters (i.e. force exerted by the subject, smoothness of the movement, etc., *Reinkensmeyer et al., 2016*). It is also increasingly recognized that physical rehabilitation should be combined with 'plasticity-stimulating' or neuromodulatory interventions that render the spared CNS networks more susceptible to experience-dependent modifications (*Zeiler and Krakauer, 2013*; *Straudi et al., 2016*; *Tran et al., 2016*; *Alia et al., 2017*).

In this context, plasticity in the injured hemisphere plays a major role in post-stroke motor recovery and is a primary target for rehabilitation therapy. Indeed stimulation of the ipsilesional motor cortex, especially when paired with motor training, facilitates plasticity and functional restoration (*Allman et al., 2016*; *Dodd et al., 2017*). On the other hand, the role of the contralesional hemisphere remains highly controversial (*Dancause et al., 2015*; *Talelli et al., 2012*; *Buetefisch, 2015*). Attempts to promote stroke recovery by inhibiting the contralesional hemisphere are based on the interhemispheric competition model, which posits an enhanced transcallosal inhibition exerted by the healthy side over the spared perilesional tissue (*Murase et al., 2004*; *Dancause et al., 2015*; *Barry et al., 2014*; *Silasi and Murphy, 2014*; *Boddington and Reynolds, 2017*; *Mansoori et al., 2014*). However, direct electrophysiological measures of the evolution of interhemispheric inhibition post-stroke are not yet available. Second, inactivation of the healthy hemisphere via either low-frequency, repetitive transcranial magnetic stimulation (rTMS) or cathodal (inhibitory) transcranial direct current stimulation (tDCS) has yielded some positive yet variable effects in clinical trials (*Kim et al., 2010*; *Lefaucheur et al., 2014*; *Di Pino et al., 2014*; *Plow et al., 2016*; *Volz et al., 2017*). This variability in outcome may depend on the extent of damage: interhemispheric competition dominates in patients with limited damage in the affected hemisphere, while after large lesions the contralesional side appears to vicariate the lost functions (*Bradnam et al., 2012*; *Di Pino et al., 2014*).

For these reasons, there is a pressing need for a deeper understanding of the basic physiology of interhemispheric interactions and their role in motor recovery (*Murase et al., 2004*; *Vallone et al., 2016*; *Boddington and Reynolds, 2017*; *Volz et al., 2017*). This requires validated animal models of stroke which allow well-controlled experimental conditions, and the possibility to perform detailed investigations about the mechanisms underlying post-stroke recovery (*Alia et al., 2017*; *Ward, 2017*). Mice and rats are widely employed for this purpose, and several quantitative measures of motor function have been developed in these species (*Corbett et al., 2017*). Indeed, the marked similarities in kinematic parameters of skilled reaching between humans and rodents indicates that mice can be used in pre-clinical studies with excellent translation potential (*Klein et al., 2012*). On the other hand, it should be noted that, unlike primates and humans, rodents do not have direct cortico-motoneuronal connections as the fastest pyramidal connections are dysinaptic (*Alstermark and Ogawa, 2004*; *Alstermark and Pettersson, 2014*).

In this manuscript, we define a rehabilitation protocol to promote 'true' recovery of motor function (i.e., the restoration of pre-lesion movement patterns) after focal cortical stroke in mice (*Lai et al., 2015*). First, we demonstrate that the healthy, contralesional hemisphere exerts an increased transcallosal inhibition over the spared perilesional tissue. To counteract such inhibitory influences, we employed a reversible inactivation of the healthy, contralesional motor cortex. Silencing of the contralesional cortex was coupled with training in a robotic device, the *M-Platform* (designed on the basis of one of the first human rehabilitation robots, the ArmGuide, *Reinkensmeyer et al., 2000*) that allows intensive and highly repeatable exercises of the mouse forelimb (*Spalletti et al., 2014*). Such combined treatment normalized transcallosal inhibition and promoted recovery in general motor tests and in manual dexterity (i.e. skilled reaching) with a remarkable restoration of pre-lesion movement patterns.

## Results

### Evolution of interhemispheric functional connectivity after stroke

A focal ischemic lesion was induced in the primary motor cortex (M1) of mice, targeting the caudal forelimb area (CFA) by means of Rose Bengal-induced phototrombosis (*Lai et al., 2015*; *Alia et al., 2016*). No damage was observed in either the rostral forelimb area (RFA) or the posterior hindlimb motor cortex (*Lai et al., 2015*). The procedure led to a complete loss of neurons in all cortical layers of the illuminated hemisphere (*Figure 1a*). The lesion was typically restricted to the gray matter (see Nissl staining in *Figure 1a*). In a minority (about 25%) of the mice, partial damage was detected in the dorsal aspect of the white matter, as demonstrated by reduced staining for myelin basic protein (MBP; *Figure 1a*, inset).

We first examined changes in interhemispheric interactions between the contralesional and ipsilesional RFAs at different times after stroke (5 and 30 days). We used optogenetic stimulation of the contralesional RFA in Thy1-ChR2 mice expressing the light-activated cation channel ChR2 mainly in

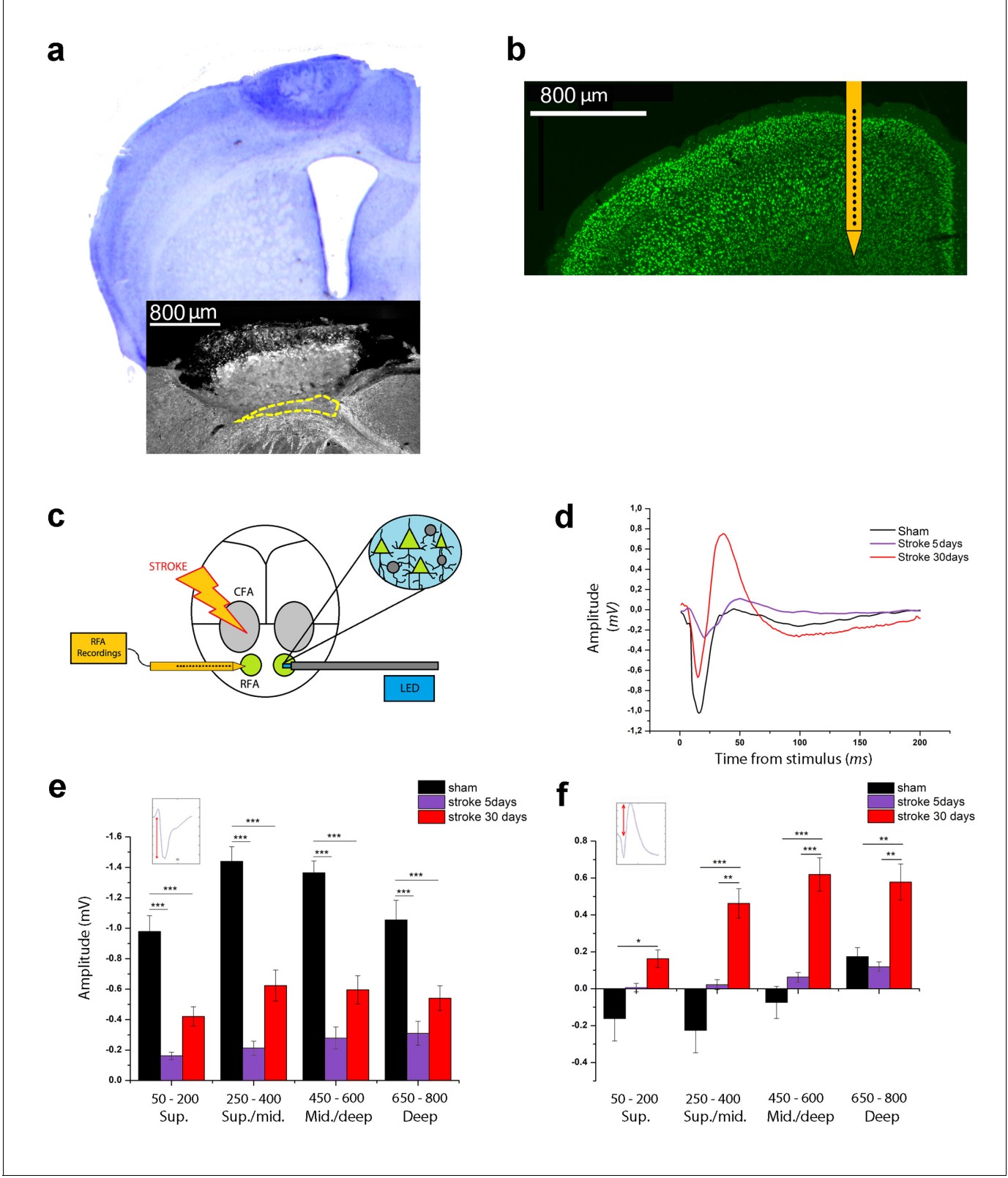

**Figure 1.** Optogenetic assessment of alterations in transcallosal connectivity between homotopic premotor areas after stroke. (a) Top: representative Nissl staining of a brain coronal section including the ischemic lesion. Bottom:example of MBP staining in a mouse with partial callosal lesion (indicated by dashed line). (b) Representative NeuN immunostaining of a coronal section indicating the position of the recording electrode in the RFA. (c) Schematic of the experimental protocol. Green triangles represent ChR2-expressing pyramidal neurons activated by blue light. (d) Representative Field

*Figure 1 continued on next page*

*Figure 1 continued*

Potential response in layer V of the RFA in sham (Black) and ischemic animals at 5 (Purple) and 30 (Red) days post stroke after single pulse stimulation in the contralateral RFA. (e, f) Mean amplitude of the early negative (e) and late positive (f) components of evoked field potentials in sham (Black) and ischemic animals at 5 (Purple) and 30 (Red) days post stroke. The inset illustrates quantification of the baseline-to-peak amplitude. The early component is smaller in ipsilesional RFA of stroke animals while the late positive waveis significantly higher at 30 days with respect to controls across all cortical layers (Two-way Anova, followed by Holm-Sidak test, *p<0.05; **p<0.01; ***p<0.001 between groups). Results from adjacent channels (depths) were pooled to show responses in superficial (Sup), Sup/Middle, Middle/deep and Deep layers. Data are mean ±SE.

DOI: https://doi.org/10.7554/eLife.28662.002

The following source data and figure supplements are available for figure 1:

**Source data 1.** Mean and SEM are presented for the data in *Figure 1*.
DOI: https://doi.org/10.7554/eLife.28662.005

**Figure supplement 1.** Pharmacological dissection and light-dependency of the evoked Field Potential.
DOI: https://doi.org/10.7554/eLife.28662.003

**Figure supplement 1—source data 1.** Mean and SEM are presented for the data in *Figure 1—figure supplement 1*.
DOI: https://doi.org/10.7554/eLife.28662.004

layer V pyramidal neurons. We recorded Local Field Potentials (LFPs) and Multi Unit Activity (MUA) in the ipsilesional RFA of ischemic and sham animals. Extracellular recordings were performed with 16-channel silicon probes (*Figure 1b,c*). In deep layers, single-pulse stimulation evoked a FP with two main components: (i) an early negative wave, peaking between 17 and 26 ms after stimulation and (ii) a late positive component, peaking between 40 and 60 ms (*Figure 1—figure supplement 1a*, black trace). Pharmacological dissection of the two components indicated that the early negative-going FP is due to direct transcallosal excitation, while the late positive peak corresponds to disynaptic inhibition of target neurons via local GABAergic cells (*Restani et al., 2009*; *Palmer et al., 2012*). Indeed, local delivery over the cortex of the GABA-B antagonist CPG55845 in naïve mice had no significant effect on the negative-going FP (*Figure 1—figure supplement 1a,b*) but substantially reduced the positive-going component (*Figure 1—figure supplement 1a,c*), consistent for a role of GABA-B signaling in interhemispheric inhibition (*Irlbacher et al., 2007*; *Palmer et al., 2012*).Topical application over the cortex of CNQX, a blocker of glutamatergic transmission, completely abolished the responses, confirming the post-synaptic origin of the recorded FPs (*Figure 1—figure supplement 1a*, blue trace). Altogether, these data demonstrate that the first phase of the evoked potential corresponds to a volley of transcallosal excitation in the RFA which leads to local activation of GABAergic cells and subsequent GABA-B dependent inhibition of cortical neurons (i.e. outward currents resulting in late positive peaks in the FPs).

We next quantified changes in the evoked FPs in sham and stroke animals (5 and 30 days after stroke) to determine the evolution of interhemispheric functional connectivity. Analysis of the initial, negative component demonstrated that the FP amplitude was dramatically dampened in stroke animals at 5 days (two way ANOVA, followed by Holm-Sidak test, p<0.05; *Figure 1d,e*), possibly due to connectional diaschisis (*Carrera and Tononi, 2014*). Despite this reduced transcallosal volley, there was a trend for enhanced amplitude of positive-going FP (*Figure 1d,f*). At 30 days after stroke, the main negative wave was partly reinstated, but still significantly lower with respect to sham controls (two way ANOVA, followed by Holm-Sidak test, p<0.05; *Figure 1d,e*). Quantification of the late positive component showed greater amplitudes in stroke animals at 30 days, with difference maximized in the central-deep layers (two way ANOVA, followed by Holm-Sidak test, p<0.05; *Figure 1f*). Altogether these data demonstrate an imbalance in interhemispheric connectivity early after stroke, with weaker direct excitation but paradoxical enhancement of transcallosally mediated inhibition.

## Reduced excitation and enhanced inhibition from the intact to the stroke hemisphere

We went on to further characterize the changes in interhemispheric interactions at 30 days after the infarct. Determination of input/output curves (i.e. amplitude of the negative-going FP vs. intensity of stimulation) established that the amplitude of the response progressively increased with light intensity in both healthy and stroke mice. However, the responses of ischemic animals were scaled down

and reached saturation earlier (n = 5 Stroke and n = 5 Sham animals, two way ANOVA, followed by Tukey test, p<0.05; *Figure 1—figure supplement 1d*).

To determine the laminar location and direction of the membrane currents underlying the evoked FPs, we performed a Current Source Density (CSD) analysis in healthy (Sham n = 5) and ischemic (Stroke n = 5) animals at 30 days (see *Figure 2a*). The analysis revealed a main current sink (red) spanning all cortical layers and with the shortest latency in infragranular layers. These initial current sinks were followed by sources (blue) which were particularly prominent in the deep layers of the ipsilesional RFA.

The current sinks were consistently dampened in the RFA of ischemic animals at 30 days (*Figure 2a*), in keeping with reduced response of recorded neurons to direct transcallosal excitation from the contralesional to the stroke side. To strengthen this conclusion we quantified MUA activity in RFA of healthy and stroke animals following stimulation of the contralateral homotopic area. The analysis revealed robust stimulus-evoked response at a depth corresponding mainly to cortical layer V in control mice (*Figure 2b*, top). This activity was substantially reduced in stroke mice (*Figure 2b*, bottom).

To confirm an asymmetry of interhemispheric communication after stroke, in a subset of ischemic mice (n = 5) we performed recordings in both RFAs after stimulation of the contralateral side. The results showed that the evoked FP in the contralesional RFA of stroke animals was remarkably distinct from the response recorded in the injured RFA (see *Figure 2c*). In particular, the negative and positive components were always higher and lower, respectively, in the contralesional RFA than in the perilesional RFA of individual stroke mice (*Figure 2d,e*; paired T test, negative component p=0.003 and positive component p=0.012). In naïve controls, the amplitude of the two FP components was comparable in the two hemispheres (*Figure 2d,e*; paired T test, negative component p=1 and positive component p=0.06). These data demonstrate directional differences in interhemispheric processing between premotor areas following a localized cortical infarct in the CFA. The robust asymmetry between the FP response recorded in the ipsilesional and contralesional cortex after transcallosal stimulation suggests that the alterations in interhemispheric communication are caused by plastic rearrangements after stroke and not by direct callosal damage.

We next focused on the stroke-induced enhancement of transcallosal inhibition in the ipsilesional side. As expected, local delivery over the cortex of the GABA-B antagonist CPG55845 potently reduced the positive-going FP 30 days after stroke (*Figure 3a,b*), while the early negative wave was slightly increased (*Figure 3—figure supplement 1*). We reasoned that the late outward, hyperpolarizing currents (see CSD analysis, *Figure 2a*) should short-circuit the excitatory input carried by a second, closely spaced stimulus leading to paired-pulse depression (PPD). Indeed, using two optogenetic stimuli delivered at Inter-Stimulus Intervals (ISIs) of 50, 100 and 200 ms, we observed in healthy animals a decrease in the magnitude of the synaptic response to the second stimulus, indicating PPD which was maximal at 50 ms (n = 5 animals, *Figure 3—figure supplement 2*). Thus we compared the Paired Pulse Ratio (PPR), that is, the ratio of the second to the first postsynaptic response at an ISI of 50 ms in control and stroke mice at 30 days (*Figure 3c*). The PPR was significantly lower in stroke animals with respect to controls, consistent with enhanced transcallosal inhibition in ischemic vs. sham animals (*Figure 3d*). The field PPR results were confirmed by performing the same analysis with MUA. As shown in *Figure 3e*, in sham animals (n = 5) we observed a depressed response to the second stimulus with respect to the first but the depression was enhanced after stroke (n = 5). In particular, the PPR was significantly lower in middle-deep layers of stroke animals (T test, p<0.05).

## Silencing of the healthy hemisphere partially improves motor outcomes after stroke

The data reported so far clearly indicate that the ischemic event perturbs the balance between the two hemispheres. To restore such a balance we performed a focused inactivation of the forelimb motor cortex in the contralesional hemisphere by means of intracortical injections of the synaptic blocker Botulinum Neurotoxin E (BoNT/E). This neurotoxin is known to block neurotransmission preferentially in excitatory terminals by SNAP-25 cleavage (*Costantin et al., 2005*; *Caleo et al., 2007*).

To evaluate the spread of toxin activity in the motor cortex, we injected a group of 5 naïve mice into the CFA and we sacrificed them 2 days post injection (*Caleo et al., 2007*; *Antonucci et al., 2010*) for tissue processing. We performed immunostaining for intact and cleaved SNAP-25

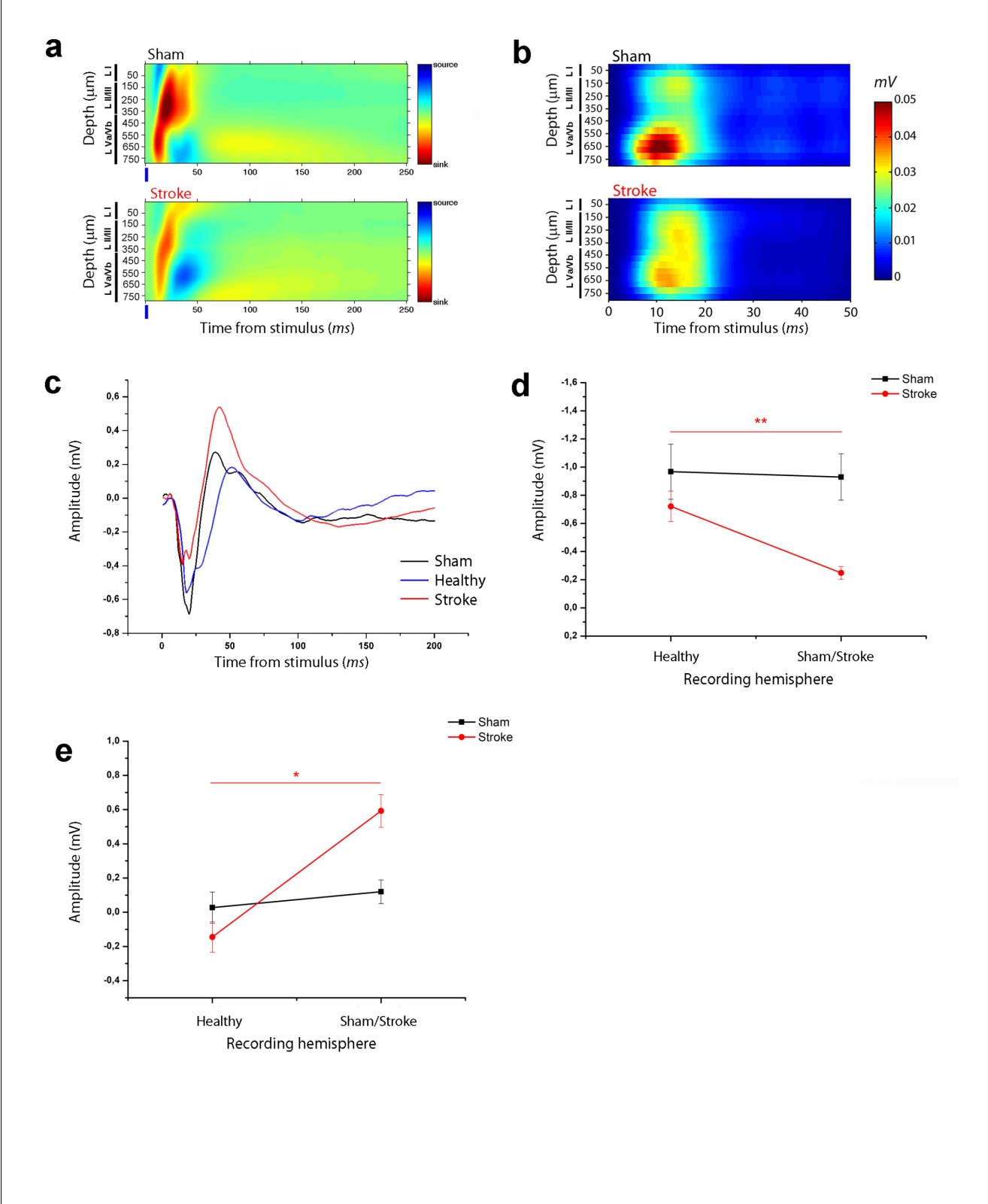

**Figure 2.** Asymmetry in functional transcallosal connectivity in stroke animals. (a) Current source density (CSD) analysis of the cortical field potential response to optogenetic stimulation, in naïve (top) and stroke mice (bottom). Warm colours (yellow and red) represent current sinks and cool colours (dark and light blue) represent current sources. Blue bars represent the light pulse. Roman numerals indicate cortical layers. (b) Multi Unit Activity (MUA) triggered by single-pulse stimulation across all cortical layers in the target hemisphere in sham (top) and stroke (bottom) mice. Please note the lower

*Figure 2 continued on next page*

*Figure 2 continued*

MUA values in infragranular layers of stroke animals with respect to controls. (c) Representative Field Potential (FP) response recorded in layer V of the ipsilesional (Red) and contralesional cortex (Blue) of a stroke animal after transcallosal stimulation. The field response in a control, naïve animal (Black trace) is reported for comparison. (d, e) Amplitude of the first negative (d) and late positive component (e) of the FP recorded in the healthy hemisphere (Healty) after stimulation of the stroke side and vice versa (Sham/Lesioned). Note the robust asymmetry in interhemispheric communication in stroke mice. Paired T test, *p<0.05; **p<0.01.

DOI: https://doi.org/10.7554/eLife.28662.006

The following source data is available for figure 2:

**Source data 1.** Mean and SEM are presented for the data in *Figure 2*.
DOI: https://doi.org/10.7554/eLife.28662.007

(*Figure 4a,c,d and b,e,f*, respectively). We found that SNAP-25 cleavage was evident along all cortical layers and spanned the entire CFA. We then characterized the duration of BoNT/E activity in the motor cortex by collecting tissue samples of the injection site at different times after surgery. Western Blot analysis indicated persistence of cleaved SNAP-25 for at least 10 days after injection (*Figure 4g*). Behavioral analysis of motor function in naïve mice (n = 5) injected unilaterally with BoNT/E showed a transient deficit in performance of the contralateral forelimb (evaluated with Gridwalk and Schallert Cylinder tests), which was significant on day two post-injection (two way RM ANOVA followed by Tukey test, Day 2 Gridwalk p=0.014, Schallert p=0.03; *Figure 4—figure supplement 1a,b*), consistent with high levels of cleaved SNAP-25 (*Figure 4g*). Performance returned to baseline on day 9 (Gridwalk p=0.251 and Schallert p=0.781) and remained stable up to 30 days post-injection (*Figure 4—figure supplement 1a,b*).

In order to assess the effect of the sole inactivation of the contralesional hemisphere, a group (n = 5, BoNT) of animals subjected to photothrombosis received, in the same surgery, intracortical infusion of BoNT/E (*Figure 5*). We assessed motor performances of these animals once a week with classical behavioural tests, Gridwalk and Schallert Cylinder test, up to 30 days post lesion. Motor performances were compared with those of untreated stroke animals (n = 11). As shown in *Figure 4h*, the silencing of the healthy hemisphere led to functional gains in the Gridwalk test but was completely ineffective in the Schallert Cylinder task. In the Gridwalk test, the BoNT/E-treated group was superior to untreated stroke at several time points (two way RM ANOVA followed by Tukey test, day 9 p=0.05, day 16 p=0.002, day 23 p=0.001). However, at 30 days the BoNT group was not distinguishable from untreated stroke (p=0.094). In the Schallert Cylinder test, the deficit measured at day two in BoNT/E-treated animals was lower than in untreated stroke (two way repeated measures ANOVA followed by Tukey test, p<0.05; *Figure 4h*, lower panel). Since the Cylinder test measures the asymmetry in the forelimb use, this acute effect is likely due to synaptic silencing of the contralesional cortex (consistent with the data shown in *Figure 4—figure supplement 1*). However, over time the deficit in performance of BoNT/E-injected mice caught up with that measured in untreated stroke controls (two way RM ANOVA followed by Tukey test, day 30, BoNT/E vs. untreated stroke, p=0.903).

## Robotic rehabilitation promotes task-specific forelimb improvements not generalized to other motor functions

We next investigated the impact of physical rehabilitation by subjecting the affected forelimb of stroke mice to daily controlled, repeatable and targeted exercises, guided by the *M-Platform* (*Figure 5c*), previously designed and characterized in our laboratory (*Spalletti et al., 2014*). This device allows head-restrained mice to perform several sessions of forelimb retraction with the possibility to monitor the motor performance and collect quantitative parameters, such as the time required to accomplish the task (t-target) or the number of attempts (i.e. the number of force peaks not resulting in a displacement of the handle). Here, we used this device to train injured mice and to evaluate whether the effect of the training could be generalized to other forelimb tasks. The acute effect of the lesion was evaluated two days after surgery with Schallert and Gridwalk tests while the daily robotic training started from day five and continued four days a week up to day 30 (n = 10 mice; *Figure 5c*). Consistent with previous results (*Spalletti et al., 2014*), the daily robotic training induced an improvement in the parameters related to the retraction task on the *M-Platform*

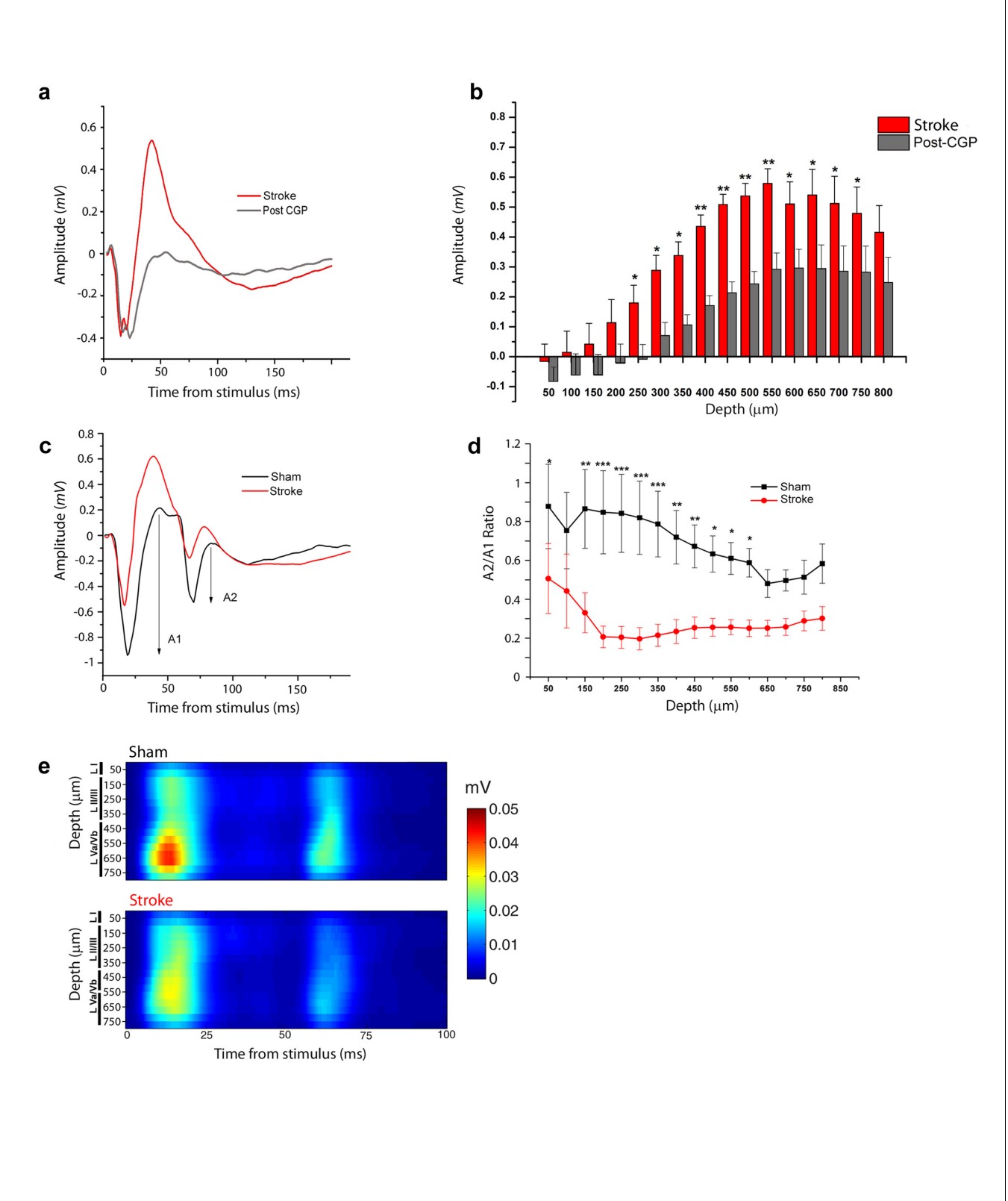

**Figure 3.** Increased interhemispheric inhibition in stroke animals. (**a**) Representative Field Potential response to single pulse stimulation in the contralateral RFA in layer V of the RFA in ischemic animals before (Red) and after (Grey) application of the GABA-B antagonist CGP 55845. (**b**) Mean amplitude of the late positive component of evoked FP before (red) and after (Grey) application of the GABA-B antagonist CGP 55845. The late positive component is significantly lower after CGP55845 application (Two-way Anova, followed by Holm-Sidak test, stroke vs Post-CGP55845, *p<0.05;
*Figure 3 continued on next page*

Figure 3 continued

**p<0.01; ***p<0.001). Data are mean ±SE. (c) Representative Field Potential response evoked by paired pulse stimulation (50 ms inter-stimulus interval) recorded in the RFA of a sham (Black) and astroke (Red) animal. A1 and A2 represent the amplitude of the first and second response, respectively. (d) Paired Pulse ratio was consistently lower in ischemic (Red) versus healthy animals (Black). ***p<0.001 (Two-way Repeated Measures Anova followed by Holm-Sidak test, sham vs stroke). Data are mean ±SE. (e) MUA triggered by 50 ms ISI paired-pulse stimulation across all cortical layers in the target hemisphere in sham (top) and stroke (bottom) mice. Please note the increased paired-pulse depression in stroke animals with respect to controls, in middle-deep layers (p<0.05, T test). Roman numerals indicate cortical layers.

DOI: https://doi.org/10.7554/eLife.28662.008

The following source data and figure supplements are available for figure 3:

**Source data 1.** Mean and SEM are presented for the data in *Figure 3*.
DOI: https://doi.org/10.7554/eLife.28662.013

**Figure supplement 1.** Effect of GABA-B blocker CGP55845 on the early negative FP.
DOI: https://doi.org/10.7554/eLife.28662.009

**Figure supplement 1—source data 1.** Mean and SEM are presented for the data in *Figure 3—figure supplement 1*.
DOI: https://doi.org/10.7554/eLife.28662.010

**Figure supplement 2.** Variation of PPD as a function of ISI.
DOI: https://doi.org/10.7554/eLife.28662.011

**Figure supplement 2—source data 1.** Mean and SEM are presented for the data in *Figure 3—figure supplement 2*.
DOI: https://doi.org/10.7554/eLife.28662.012

(*Figure 6—figure supplement 1a,b*). However, we found that this improvement was not generalized to other forelimb task, as demonstrated by the lack of recovery in Schallert and Gridwalk tests, where the motor deficit remained stable over the observation period (*Figure 6a,b*).

## Robotic training combined with transient inactivation of contralesional hemisphere triggers 'true' motor recovery

We finally tested a combined approach by coupling daily robotic rehabilitation with BoNT/E injection into the contralesional hemisphere (*Figure 5d*). In *Figure 6a,b*, motor performances in the Gridwalk and the Schallert Cylinder tests are reported for Robot and Robot+BoNT (n = 11) groups. In both tests, Robot+BoNT mice showed significant improvements, especially at day 30 (two way repeated measures ANOVA followed by Tukey test, Robot+BoNT vs Robot, Gridwalk p<0.001, Schallert p<0.05).

It was also important to check whether the positive effects of the combined rehabilitation last beyond the treatment period. To this aim, two additional cohorts of Robot+BoNT and Robot only mice (n = 5 per group) were tested at 40 days, that is, after a follow-up period of 10 days during which robotic training was suspended. We found that performances in Robot+BoNT group remained improved at 40 days with respect to animals undergoing only robotic rehabilitation (two way repeated measures ANOVA, Robot vs. Robot+BoNT 40 days, p<0.05) (*Figure 6c,d*). Of note, no differences between 30 and 40 days time points were found for both groups. These findings indicate functional gains persisting beyond the window of treatment.

To directly compare the effectiveness of the combined treatment vs. either therapy alone, we plotted the motor performance of all the treated animals at the completion of training (30 days, *Figure 7a,b*). In the Gridwalk test, Robot+BoNT (n = 11) group was superior to Robot (n = 10) and untreated stroke (one way ANOVA followed by Tukey test p<0.001), but statistically comparable to BoNT/E only (p=0.45). However, while the Robot+BoNT is different from untreated stroke, this was not true for both Robot (p=0.79) and BoNT (p=0.175) groups. In the Schallert cylinder test, mice receiving the combined therapy displayed markedly improved performances with respect to all the other groups (one way ANOVA followed by Tukey test, p=0.023 vs BoNT, p=0.005 vs stroke untreated and p=0.01 vs Robot). Altogether, these data indicate that the combined treatment yields better results that the separate therapies.

To further characterize the impact of the combined therapy, animals were also tested in the Single Pellet Retrieval task (*Lai et al., 2015*). They were pre-trained during 2–3 weeks post-stroke and then tested weekly after the infarct to examine (i) the percentage of correct graspings and (ii) the kinematics of reaching. Robot+BoNT animals (n = 5) were compared to untreated stroke mice (n = 6). We found that the percentage of errors in the Single Pellet Retrieval task increased 2 days

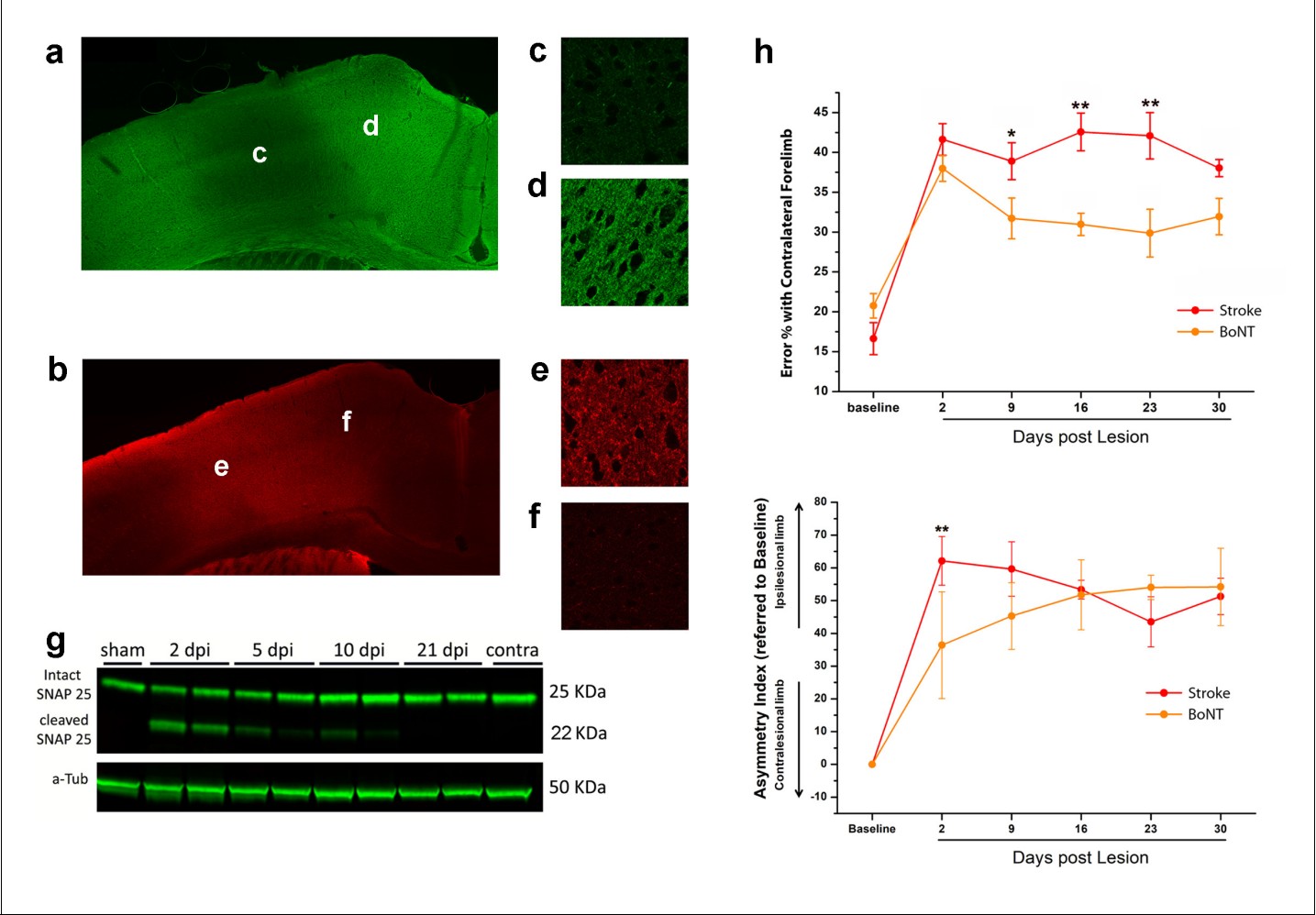

**Figure 4.** BoNT/E action in motor cortex and its impact on functional recovery. Immunofluorescence staining for intact (**a**) and cleaved (**b**) SNAP-25. Magnification of the injection site showed the nearly total absence of intact SNAP-25 (**c**) whereas the cleaved SNAP-25 signal is consistent (**e**). Conversely, in the peri-injection site there is a conspicuous amount of intact SNAP-25 (**d**), but no signal from cleaved SNAP-25 (**f**). (**g**) Representative immunoblotting for intact (25 KDa) and cleaved (22 KDa) SNAP-25 on cortical protein extracts from injected mice. Tissues were harvested from the treated area at different days post-injection (dpi 2, 5, 10 and 21), from the motor cortex contralateral to the injection side (contra) and from a control animal (sham). Each lane represents one animal. a-Tub means α-Tubulin (internal standard). (**h**) Pre- and post-lesion performance of the stroke untreated (Red) and BoNT/E injected (Orange) groups measured as the percentage of contralesional forelimb foot faults in the Gridwalk task (upper panel) and as Asymmetry Index in the SchallertCylinder test (lower panel). *p<0.05, **p<0.01 vs stroke untreated (Two-way Repeated Measures Anova, followed by Holm-Sidak test).

DOI: https://doi.org/10.7554/eLife.28662.014

The following source data and figure supplements are available for figure 4:

**Source data 1.** Mean and SEM are presented for the data in *Figure 4*.
DOI: https://doi.org/10.7554/eLife.28662.017

**Figure supplement 1.** BoNT injection in Sham animals induces a transient deficit in performance of the injured forelimb.
DOI: https://doi.org/10.7554/eLife.28662.015

**Figure supplement 1—source data 1.** Mean and SEM are presented for the data in *Figure 4—figure supplement 1*.
DOI: https://doi.org/10.7554/eLife.28662.016

after stroke but was recovered in the animals with combined therapy at 16 days and remained stable at 30 days (Two way RM ANOVA followed by Tukey test vs Baseline, p=0.55 at day 16 and p=0.989 at day 30; *Figure 8—figure supplement 1*). Recovery of prehension was significantly more robust in the Robot+BoNT group as compared to untreated stroke (p<0.01; *Figure 8—figure supplement 1*).

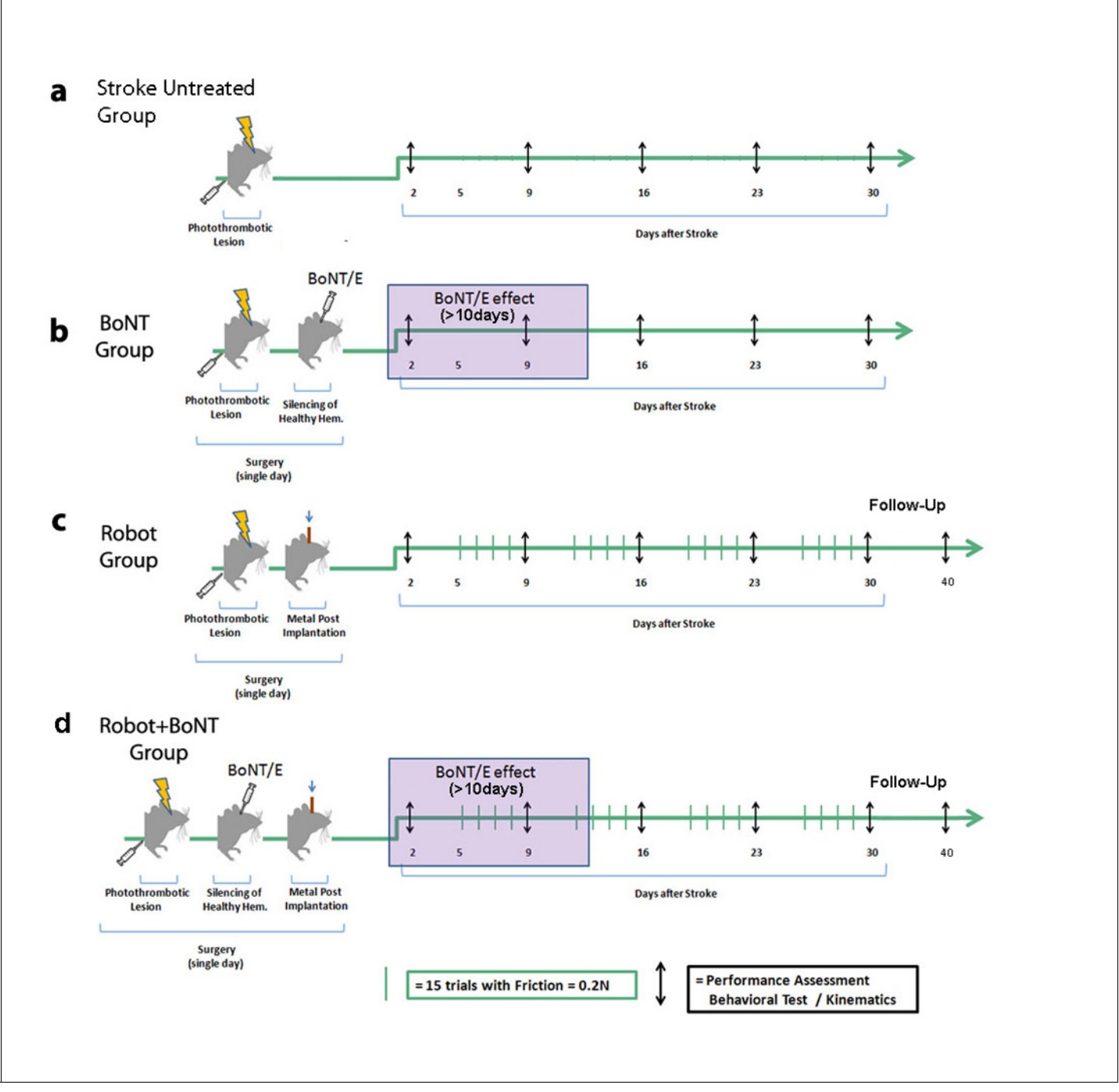

**Figure 5.** Schematic of the treatment protocols. Baseline performances in behavioral tests were assessed for all groups before the surgery and then once a week up to 30 days post lesion (black arrows). Stroke animals with no treatment were included in each experimental cohort (**a**). In the BoNT group (**b**), photothrombosis was immediately followed by BoNT/E injection into the contralesional hemisphere with toxin effect lasting >10 days (purple box). In the Robot group (**c**), the metal post for head fixation on the robotic platform was applied immediately after the photothrombotic lesion. Animals started the daily robotic training (green bars) from day 5, 4 days per week up to 30 days post lesion. In the Robot + BoNT group (**d**), mice received BoNT/E injection into the contralesional hemisphere and head-post application during the stroke surgery, and were subjected to robotic rehabilitation as the Robot group. A subset of animals in the Robot and Robot + BoNT group were also tested 40 days post-stroke to probe the persistence of the therapeutic effects (follow-up phase with no treatment).
DOI: https://doi.org/10.7554/eLife.28662.018

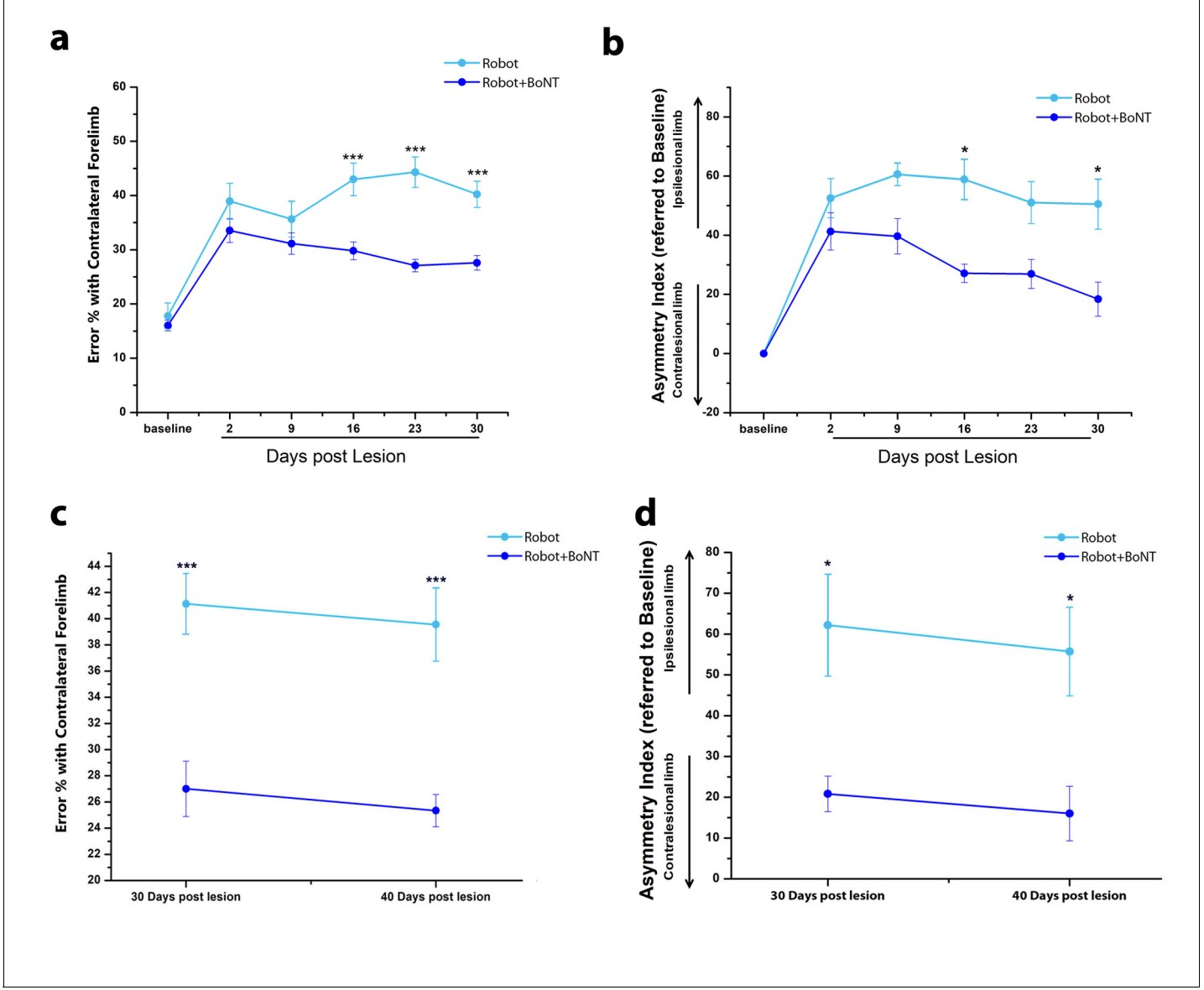

**Figure 6.** The combined therapy induces functional improvements that outlast the window of treatment. (a) Pre- and post-lesion performance of Robot (Light Blue) and Robot+BoNT (Blue) groups measured as the percentage of contralesional forelimb foot faults in the Gridwalk task. ***$p<0.001$ (Two-way Repeated Measures Anova, followed by Tukey test, Robot vs Robot+BoNT). (b) Pre- and post-lesion performance of Robot (Light Blue) and Robot+BoNT (Blue) groups measured as Asymmetry Index in the Schallert Cylinder test. ***$p<0.001$ (Two-way Anova followed by Tukey test, Robot vs Robot+BoNT). (c, d) Maintenance of the motor performances in the Gridwalk (c) and Schallert Cylinder (d) tests after 10 days of Follow-Up with no treatment. The Robot+BoNT group (Blue) remains significantly different from Robot (Light Blue) at day 40 (Two-way Repeated Measures Anova, followed by Tukey test between groups, Gridwalk $p<0.001$, Schallert Cylinder $p<0.05$).

DOI: https://doi.org/10.7554/eLife.28662.019

The following source data and figure supplements are available for figure 6:

**Source data 1.** Mean and SEM are presented for the data in *Figure 6*.
DOI: https://doi.org/10.7554/eLife.28662.022

**Figure supplement 1.** Motor performance on the robotic platform.
DOI: https://doi.org/10.7554/eLife.28662.020

**Figure supplement 1—source data 1.** Mean and SEM are presented for the data in *Figure 6—figure supplement 1*.
DOI: https://doi.org/10.7554/eLife.28662.021

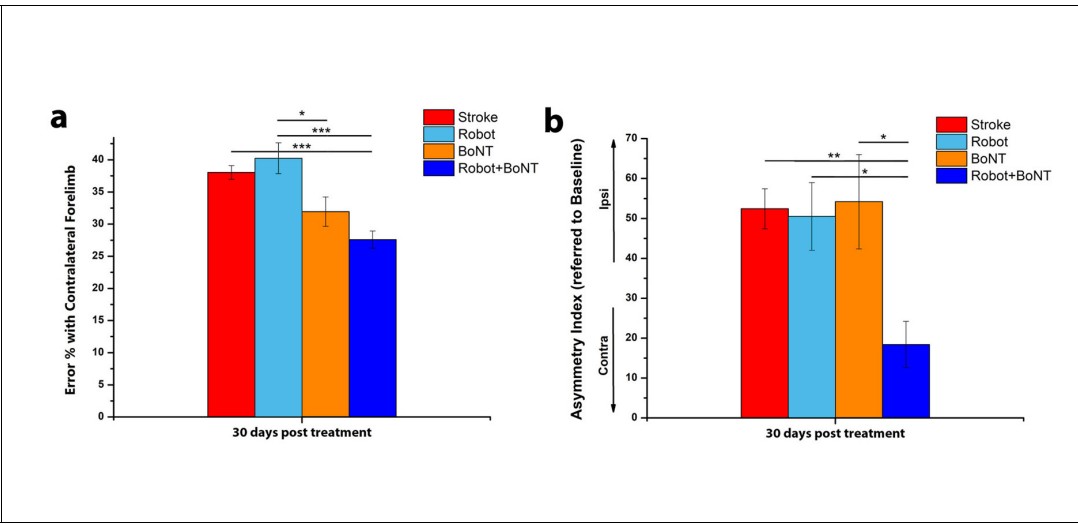

**Figure 7.** The combined treatment produces better results than the separate therapies. Percentage of contralesional forelimb foot faults in the Gridwalk task (**a**) and Asymmetry Index in the Schallert Cylinder test (**b**) at 30 days post lesion for stroke untreated (Red), Robot (Light Blue), BoNT (Orange) and Robot+BoNT (Blue). One-Way Anova followed by Tukey Test, *p<0.05, **p<0.01, ***p<0.001.

DOI: https://doi.org/10.7554/eLife.28662.023

The following source data is available for figure 7:

**Source data 1.** Mean and SEM are presented for the data in *Figure 7*.
DOI: https://doi.org/10.7554/eLife.28662.024

More importantly, the motor recovery in the Robot+BoNT mice was accompanied by the normalization of several kinematic parameters of reaching (illustrated in *Figure 8a and d*). In particular, the total length of the trajectory (ArcLen) appeared to recover at day 23 post lesion in Robot+BoNT group (*Figure 8b*, two way ANOVA followed by Tukey's test vs Baseline, p=0.08). A significant improvement was also reported at day 30 for the total area spanned by the trajectory (AUC) (*Figure 8c*, p=0.49). Of note, a significant restoration of pre-lesion values was also detected for the mean speed and for the smoothness of the reaching movement (*Figure 8e and f*, p=0.31 at day 16 and p=0.17 at Day 30 for Mean Speed and Smoothness respectively). ArcLen, AUC and mean speed measured at 30 days in the Robot+BoNT group were also significantly different from the untreated stroke group (two way repeated measures ANOVA followed by Tukey test, p<0.001). Importantly, improvements in kinematic parameters gradually emerged over time suggesting an interaction between training and contralesional cortex inactivation. Differences between the treated and control animals were not due to different extent of lesion, as the deficits on day two were equivalent (*Figure 8*) and the ischemic volumes were superimposable between the groups (T test p=0.702, *Figure 8—figure supplement 2*).

Overall these data demonstrate a remarkable synergic effect of physical robotic rehabilitation and healthy hemisphere silencing in restoring pre-lesion forelimb movement patterns.

## The combined treatment reduces the expression of 'plasticity brakes' in the perilesional tissue

To identify potential mechanisms underlying recovery, we carried out an immunohistochemical analysis of plasticity markers in the perilesional cortex of stroke untreated and Robot+BoNT groups. We performed staining for GABAergic interneurons (Somatostatin and parvalbumin-positive, SOM+ and PV+ cells) and Myelin Basic Protein (MBP), two well known plasticity 'brakes' (*Bavelier et al., 2010*), at 30 days post lesion. We found that the density of PV+ cells were decreased after stroke (n = 8) with respect to sham condition (n = 7,T test, p<0.001). The Robot+BoNT treatment further accentuated this decline and the number of PV+ cells was significantly lower with respect to untreated stroke (n = 6, *Figure 9a,b*; T test, p=0.045). The same trend was found for another important class of inhibitory interneurons, the SOM+ population. The number of these cells is significantly lower after stroke with respect to sham (n = 6 sham and n = 4 stroke untreated; T test, p<0.001) but in the

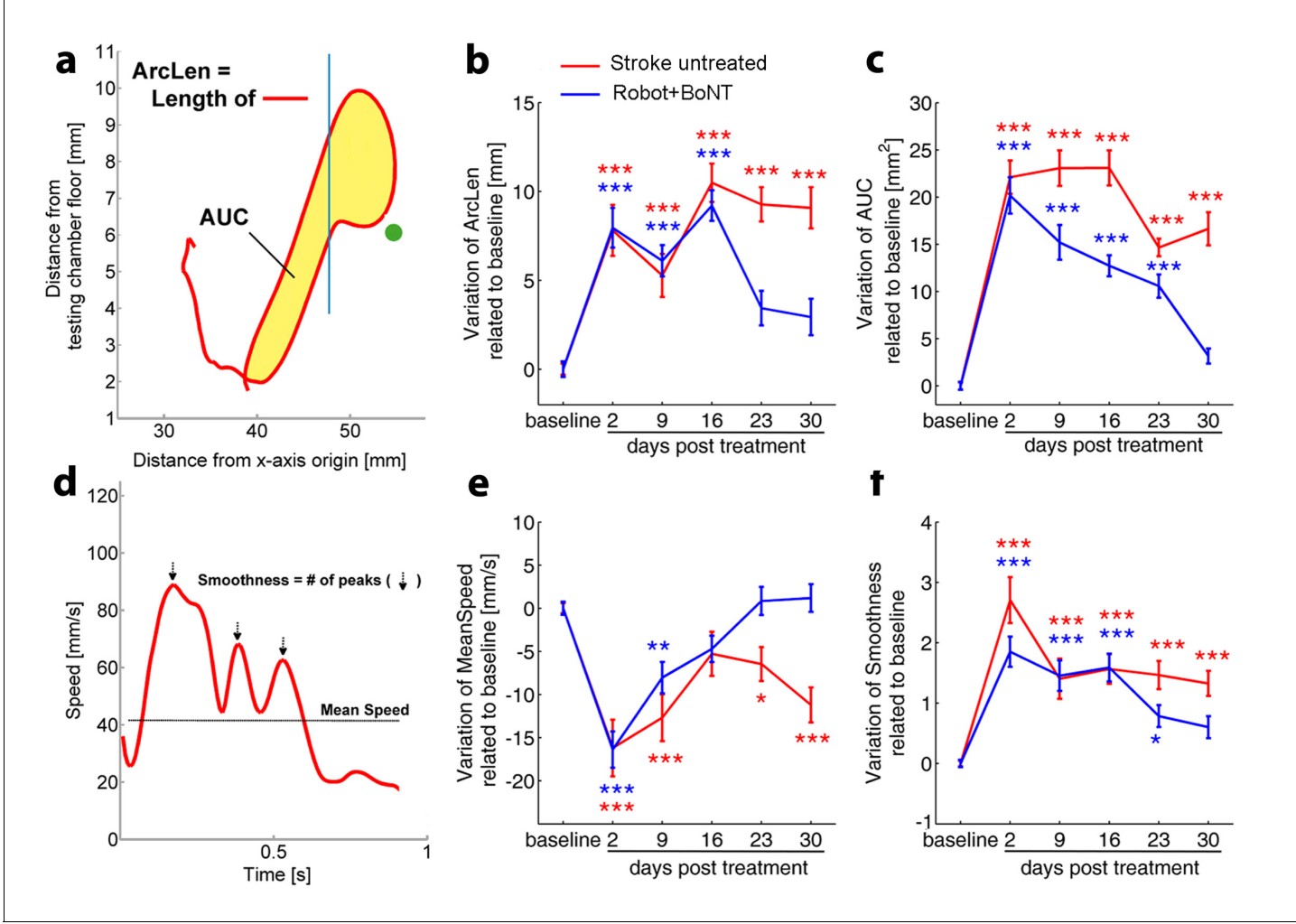

**Figure 8.** Kinematic analysis of reaching following the combined treatment. (**a**) Schematic representation of the reaching trajectory showing pellet position (green dot) and the frontal aperture of the testing cage (Blue line). AUC, *Area Under the Curve*; ArcLen, length of the curve. Longitudinal variation of *ArcLen* (**b**) and AUC (**c**) computed from reaching movement trajectories in untreated stroke (Red) and Robot+BoNT (Blue) groups. (**d**) Representative speed profile during reaching. Mean Speed and Smoothness of movement are indicated (Black line and arrows). Longitudinal variation of *Smoothness* (**e**) and *Mean Speed* (**f**) computed from reaching movement trajectories in untreated stroke (Red) and Robot+BoNT (Blue) groups. Note the substantial amelioration of the parameters in the combined treatment group. Values are normalized by subtracting baselines and plotted as the means ± standard error.Two way repeated measures ANOVA followed by Tukey test vs Baseline *p<0.05, **p<0.01, ***p<0.001.

DOI: https://doi.org/10.7554/eLife.28662.025

The following source data and figure supplements are available for figure 8:

**Source data 1.** Mean and SEM are presented for the data in *Figure 8*1.

DOI: https://doi.org/10.7554/eLife.28662.030

**Figure supplement 1.** Combined treatment significantly improved motor function in end-point analysis of a skilled task.

DOI: https://doi.org/10.7554/eLife.28662.026

**Figure supplement 1—source data 1.** Mean and SEM are presented for the data in *Figure 8—figure supplement 1*.

DOI: https://doi.org/10.7554/eLife.28662.027

**Figure supplement 2.** Volumes of the ischemic lesion are statistically comparable in untreated stroke and Robot+BoNT groups.

DOI: https://doi.org/10.7554/eLife.28662.028

**Figure supplement 2—source data 1.** Mean and SEM are presented for the data in *Figure 8—figure supplement 2*.

DOI: https://doi.org/10.7554/eLife.28662.029

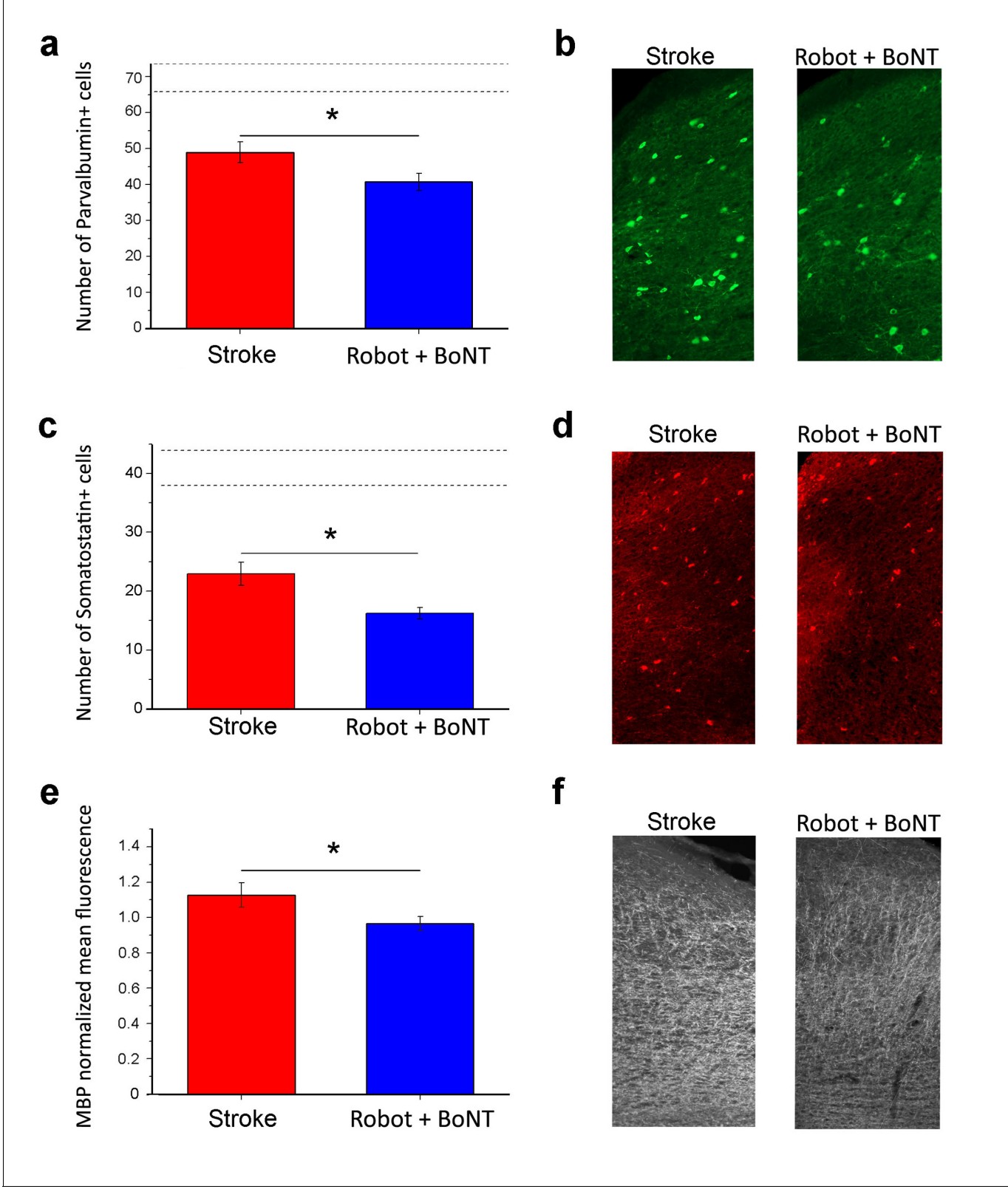

**Figure 9.** Reduction of plasticity 'brakes' after combined treatment. The number of PV+ cells decreased with combined treatment (**a**) as shown by representative micrographs of the counting region (**b**; Robot+BoNT n = 6, untreated stroke n = 8). Consistently, the number of SOM+ cells in the perilesional tissue was also decreased after combined therapy (**c**) and **d**), Robot+BoNT n = 6, untreated stroke n = 4). The mean fluorescence of MBP

*Figure 9 continued on next page*

*Figure 9 continued*
staining indicated that the rehabilitated animals (n = 4) have less MBP in the perilesional cortex with respect to non-rehabilitated group (n = 3) (**e** and **f**). In **a** and **c**, the dotted lines indicate the range of values for naïve, unlesioned mice. Data are mean ±SE. (T test *p<0.05).
DOI: https://doi.org/10.7554/eLife.28662.031
The following source data is available for figure 9:

**Source data 1.** Mean and SEM are presented for the data in *Figure 9*.
DOI: https://doi.org/10.7554/eLife.28662.032

rehabilitated group was even more diminished (n = 6; *Figure 9c,d*; T test, p=0.014). Another important marker of cortical plasticity is MBP, one of the major myelin components (*Bartholdi and Schwab, 1998*; *McGee et al., 2005*; *Kim et al., 2009*). We compared MBP expression between stroke untreated (n = 3) and Robot+BoNT (n = 4) groups 30 days after injury. We found a significant MBP decrease in the combined treatment group (*Figure 9e,f*; T test, p=0.045), pointing to a more plastic state of the perilesional tissue in the rehabilitated animals.

## Reduction of transcallosal inhibition in the combined treatment group

We finally tested whether the combined therapy affects the enhanced interhemispheric inhibition following ischemia. We added a second Robot+BoNT group of Thy1-ChR2 transgenic mice (n = 5) and we verified the functional recovery with the Gridwalk test (*Figure 10—figure supplement 1*, two way ANOVA followed by Tukey's test vs Baseline, Day 23 p=0.05, Day 30 p=0.06). At the end of the treatment (30 days post-lesion), we recorded in the perilesional RFA FPs evoked by optogenetic stimulation of the contralesional cortex (*Figure 10a*). Both the CSD and the LFP quantification show that the early negative component of the FP (reflecting direct transcallosal excitation) was not rescued in the Robot+BoNT mice and only partly recovered in deep layers (two way ANOVA followed by Tukey's test p=0.089, *Figure 10b,c*). However, the late positive component (indicating transcallosal, GABA-B mediated inhibition; see *Figure 3a,b*) was almost completely normalized in the combined therapy group. In particular, the amplitude of the positive-going FP was significantly reduced with respect to untreated stroke in superficial/middle and middle/deep layers (two way ANOVA followed by Tukey's test, p=0.009). Moreover, it was comparable to sham animals across all the cortical layers (*Figure 10b,d*). In keeping with these data, also the PPD was rescued in Robot+BoNT mice in superficial/middle, middle/deep and deep layers, as shown from the PPD ratio of the field potential (two way ANOVA followed by Tukey's test, Robot+Bont vs Stroke untreated, p<0.01, *Figure 10e*). This recovery was appreciable also in the MUA analysis (*Figure 10f*). Altogether these data indicate a selective normalization of inter-hemispheric inhibition in stroke animals with combined therapy.

## Discussion

In this manuscript we induced a unilateral ischemic injury in forelimb mouse motor cortex to demonstrate an excessive transcallosal inhibition exerted by the homotopic contralesional regions over the perilesional spared motor areas. Based on these evidences, we designed a combined treatment involving transient inactivation of the healthy hemisphere and intensive sessions of exercises guided by a robotic device. This combinatorial strategy was remarkably effective to improve general forelimb motor function in tasks and, importantly, to restore pre-lesion motor patterns in a skilled forelimb test, thus avoiding the development of compensatory, possibly maladaptive, motor strategies. These ameliorations in functional outcomes were accompanied by a significant reduction in interhemispheric inhibition and by downregulation of specific 'plasticity brakes' in perilesional tissue, such as GABAergic markers, confirming a treatment-induced cortical disinhibition.

Data from both humans and animal models demonstrate that the ischemic event is followed by extensive changes in cortical excitability involving not only the perilesional tissue but also other connected brain areas in the same hemisphere or in the contralesional one (*Marshall et al., 2000*; *Ward et al., 2003*; *Murphy and Corbett, 2009*; *Clarkson et al., 2010*; *Carmichael, 2012*; *Dijkhuizen et al., 2012*; *Vallone et al., 2016*). In this context, the role of the healthy hemisphere in post-stroke recovery remains controversial (*Caleo, 2015*; *Dancause et al., 2015*). Recent evidences propose a direct relationship between the extent of the lesion and the role of the healthy

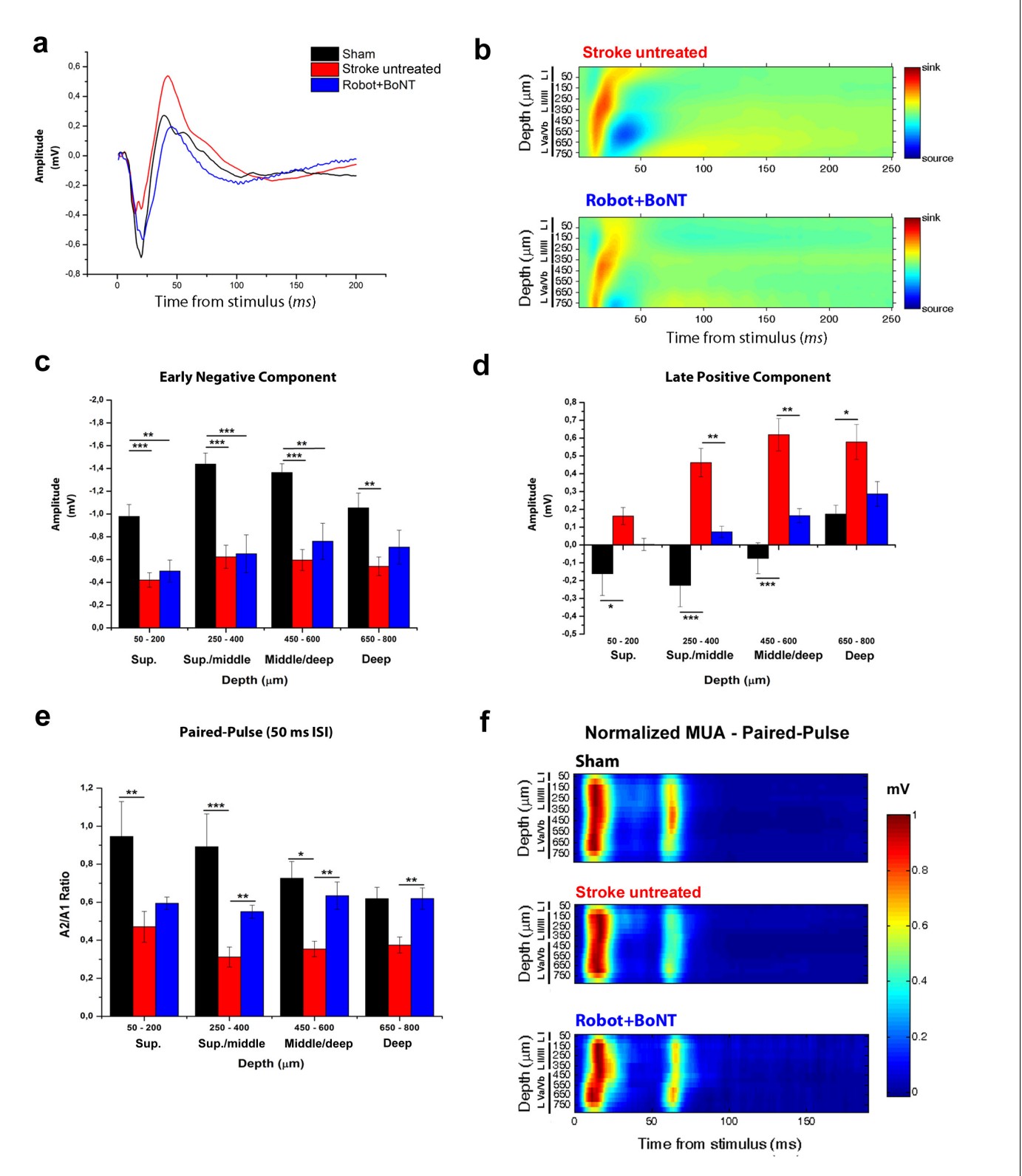

**Figure 10.** The combined treatment reduced the interhemispheric inhibition in injured animals 30 days after stroke. (**a**) Representative Field Potential response in layer V of the RFA in sham (Black), stroke untreated (Red) and Robot+BoNT (Blue) animals after single pulse stimulation in the contralateral RFA. (**b**) Current source density analysis of the cortical field potential response to optogenetic stimulation, in stroke untreated (top) and Robot+BoNT (bottom) groups. Warm colors (yellow and red) represent current sinks and cool colors (dark and light Blue) represent current sources. (**c, d**) Mean

*Figure 10 continued on next page*

*Figure 10 continued*

amplitude of the early negative (c) and late positive (d) components of evoked field potentials in sham (Black), stroke untreated (Red) and Robot+BoNT (Blue) group. Results from adjacent channels (depths) were pooled to show responses in superficial (Sup), Sup/Middle, Middle/deep and Deep layers. The early FP component in the ipsilesional RFA of rehabilitated animals is comparable to sham only in layer V (p=0.089), while the late positive wave is comparable to controls in superficial/middle to deep layers (sup p=0.436, sup./middle p=0.07, middle/deep p=0.179, deep p=0.676) and significantly lower with respect to stroke untreated in middle layers (sup./middle p=0.009, middle/deep p=0.002, Two-way Anova, followed by Tukey test between groups). (e) Paired Pulse ratio in Robot+BoNT(Blue) was comparable to sham across all the cortical layers (sup p=0.659, sup./middle p=0.267, middle/deep p=0.164, deep p=0.197, Two-way Anova, followed by Tukey test between groups) and significantly different from stroke untreated except from superficial layers (sup./middle p=0.006, middle/deep p=0.001, deep p=0.005, Two-way Anova, followed by Tukey test, Robot+BoNT vs Stroke untreated). *p<0.05; **p<0.01; ***p<0.001, data are mean ±SE. (f) MUA triggered by 50 ms ISI paired-pulse stimulation across all cortical layers in the target hemisphere in sham (top), stroke untreated (middle) and Robot+BoNT (bottom) mice. MUA values are normalized to the first peak response value, for each channel separately. Roman numerals indicate cortical layers.

DOI: https://doi.org/10.7554/eLife.28662.033

The following source data and figure supplements are available for figure 10:

**Source data 1.** Mean and SEM are presented for the data in *Figure 10*.

DOI: https://doi.org/10.7554/eLife.28662.036

**Figure supplement 1.** Functional effect of the combined Robot+BoNT treatment in ChR2 transgenic mice.

DOI: https://doi.org/10.7554/eLife.28662.034

**Figure supplement 1—source data 1.** Mean and SEM are presented for the data in *Figure 10—figure supplement 1*.

DOI: https://doi.org/10.7554/eLife.28662.035

hemisphere (*Grefkes and Ward, 2014*; *Di Pino et al., 2014*). In particular, following small strokes it is generally believed that loss of the mutual interhemispheric control leads to an excessive inhibition exerted by the undamaged hemisphere over the spared perilesional tissue (*Biernaskie et al., 2004*; *Nowak et al., 2009*; *Silasi and Murphy, 2014*; *Boddington and Reynolds, 2017*). To firmly set this point, we used a well characterized mouse model of focal ischemic injury (*Lai et al., 2015*; *Vallone et al., 2016*) to study neural mechanisms underlying interhemispheric inhibition in a standardized and controlled way. We used optogenetic stimulation to characterize transcallosal transmission between the two homotopic RFAs. We found that the first phase of the evoked potential corresponds to a volley of transcallosal excitation in the RFA, which drives local GABAergic neurons with consequent GABA-B dependent inhibition of cortical neurons (corresponding to the late positive peak in the FP). These data are consistent with neuroanatomical observations indicating that the corpus callosum consists almost entirely of excitatory fibres, which implies that interhemispheric inhibition arises from the activation of local interneurons (*Palmer et al., 2012*). We went on to demonstrate that stroke affects interhemispheric transmission, starting from the acute phase. In particular, the negative, depolarizing volley of the FP was reduced while the late positive component was enhanced. The reduction in the negative-going FP and stimulus-induced spiking after stroke can be simply explained by a dampened excitatory drive of pyramidal neurons in the ipsilesional hemisphere. The increased inhibition is consistent with the significant enhancement of PPD in stroke animals, pointing to a stronger post-synaptic GABAergic inhibition that short-circuits the excitatory input carried by a second stimulation (*Rozas et al., 2001*). It is well established that transcallosal neurons provide interhemispheric inhibition over the target hemisphere by contacting local interneurons, such as neurogliaform (NGF) cells in layer I. NGF cells are known to exert GABAergic inhibitory effects over the dendrites of layer V pyramidal neurons, as demonstrated in somatosensory cortex (*Palmer et al., 2012*). Two non-mutually exclusive possibilities may explain the increased interhemispheric inhibition following stroke: (i) a structural reorganization of callosal fibres, with enhanced innervation of NGF cells and, possibly, a retraction of inputs onto the dendrites of local pyramidal cells; (ii) an enhanced synaptic transmission between transcallosally-driven interneurons and cortical pyramids. In keeping with this idea, previous studies have shown a robust and early upregulation of GABA-B receptors in peri-infarct areas which suggest augmented responsiveness to GABA release (*Que et al., 1999*).

Based on the results obtained in optogenetic experiments, we decided to test the effects of a transient inactivation of the healthy hemisphere on general forelimb function. We used BoNT/E to block neurotransmitter release and synaptic transmission by cleaving SNAP-25, a main component of the SNARE complex (*Costantin et al., 2005*; *Caleo et al., 2007*). This approach resulted in

slightly improved outcomes in general forelimb motor tests (Gridwalk and Schallert Cylinder test) but its effect was not stable throughout our observation period. One interpretation is that silencing of the contralesional motor area stimulates plastic reorganization in perilesional tissue, but the enhancement of plasticity without the guide of an appropriate motor rehabilitation regime is not sufficient to achieve a complete recovery (*García-Alías et al., 2009*).

In the last years, robotics has received an increasing attention as a promising tool for improving repeatability, motivation and data collection in clinical stroke neurorehabilitation (*Lo et al., 2010*). Indeed, these devices offer the possibility to provide highly repeatable and customable sessions of exercises and to obtain quantitative information about the kinetics and the kinematics of the motor performance (*Caleo, 2015*). The latest randomized studies have demonstrated that robotic therapy is at least as effective as intensive conventional rehabilitation to improve motor functions, even in chronic patients with moderate-to-severe paresis (*Klamroth-Marganska et al., 2014*), and thanks to its advantages relative to costs and amount of work it is successfully used also in low- and middle-income countries (*Wagner et al., 2011*; *Bustamante Valles et al., 2016*). Here we employed the *M-Platform*, a mechatronic device for mouse forelimb training (*Spalletti et al., 2014*) that mimics a robot for upper limb rehabilitation in humans, the 'Arm-Guide' (*Reinkensmeyer et al., 2000*). Consistent with our previous results (*Spalletti et al., 2014*), daily training on the robot restores task-related parameters to pre-lesion values, indicating a specific motor recovery. However, robotic rehabilitation per se was not able to generalize recovery to untrained motor tasks (Gridwalk and Schallert Cylinder test). This indicates that the execution of daily controlled, repeatable and targeted exercises with the affected limb ameliorates strength and motor control but only for those movements that are practiced. This is consistent with previous results in clinical literature, showing only task-specific improvements with little generalization after robotic training (*Kwakkel et al., 2008*; *Panarese et al., 2012*). For these reasons, we coupled the BoNT/E-mediated suppression of activity in contralesional hemisphere with daily robotic training.

Timing of the therapeutic treatments is known to be critical for functional post-stroke outcome (*Wahl et al., 2014*). The early imbalance in interhemispheric connectivity following stroke (*Figure 1*) suggested us a prompt inactivation of the contralesional cortex via BoNT/E, which requires at least 24 hr to become fully active (*Antonucci et al., 2008*). For robotic training, we chose to start the physiotherapy at day 5, since it is known that early exercise may exacerbate brain injury, via excitotoxicity and generation of reactive oxygen species (*Kozlowski et al., 1996*; *Li et al., 2017*). Moreover, it is widely believed that priming the brain with a 'plasticity-stimulating' treatment before the beginning of a rehabilitative treatment improves motor performance after brain lesions (*Schäbitz et al., 2004*; *Starkey et al., 2012*; *Wahl et al., 2014*). Based on these considerations, BoNT/E was delivered at the time of stroke while robotic training was started 5 days later in the combined therapy group (see *Figure 5*).

Remarkably, this combined strategy was able to generalize the effects of robotic training with significant improvements in general motor coordination in Gridwalk and Schallert Cylinder tests. Improvements outlasted the window of treatment as indicated by the stable follow-up outcomes (see *Figure 6c,d*). We also evaluated the effect of the combined therapy in the Skilled Reaching test, a cortically dependent task (*Starkey et al., 2012*) that requires a fine forelimb motor control. We already demonstrated a spontaneous post-stroke amelioration in the percentage of exact reachings but not in the kinematic parameters of movement extracted by a custom-made algorithm (*Lai et al., 2015*). These results clearly point to the development of compensatory motor strategies during post-stroke evolution (*Caleo, 2015*). We found that the combined treatment improves skilled forelimb abilities, not only in terms of end-point measures but, more importantly, in terms of kinematic parameters. Indeed the speed, smoothness and shape of the reaching trajectories significantly recovered towards pre-lesion baseline. Importantly, we demonstrated that the combined treatment is able to significantly decrease the excessive interhemispheric inhibition from the healthy homotopic area over the perilesional tissue. Indeed, the late positive component of the FP, corresponding to outward, hyperpolarizing currents in the CSD, was reduced in rehabilitated animals across all the cortical layers. Moreover, analysis of PPD confirmed the normalization of interhemispheric GABAergic inhibition after the combined rehabilitative treatment. Accordingly, in rehabilitated animals we also found an increased cortical plasticity in perilesional tissue, as shown by the diminished expression of at least three important plasticity 'brakes' such as SOM+ and PV+ inhibitory interneurons and MBP (*Bavelier et al., 2010*). The decrease in PV+ and SOM+ interneurons could be due either to cell

degeneration or to a down-regulation of their markers, potentially related to an impairment in their normal physiology.In particular, the decrease in GABAergic interneurons is compatible with a local disinhibition induced by the treatment. Reduction of these markers has already been associated with improved motor outcomes triggered by motor rehabilitation (*Zeiler et al., 2013*) and to the reopening of cortical plasticity that allows network reorganization and restoration of motor function (*Ng et al., 2015*).

Coupling physical rehabilitation with non-invasive brain stimulation (NIBS) techniques has been amply exploited in the clinical practice (*Allman et al., 2016*; *Straudi et al., 2016*; *Yozbatiran et al., 2016*; *Chisari et al., 2014*). This approach has been tested in the context of constraint-induced movement therapy (*Bolognini et al., 2011*) and robotic rehabilitation but with mixed results (*Hesse et al., 2011*; *Giacobbe et al., 2013*), possibly depending on stroke size and different timings/protocols used. Studies in patients have highlighted a significant but generally small effect of physical rehabilitation combined with inhibitory NIBS applied to the contralesional cortex (*Lefaucheur et al., 2014*; *Tedesco Triccas et al., 2016*). One possible explanation for the remarkable effectiveness of robotics combined with BoNT/E silencing in the present study may be the sustained and focal inhibition of activity in the healthy hemisphere which allows a tonic downregulation of inhibitory transcallosal input. Along this line, portable and implantable devices capable of delivering continuous tDCS in stroke patients might be envisaged.

The effect of our combined treatment was not limited to task-specific motor performance but generalized to other and skilled forelimb tests. This is a key result for potential application of a combined rehabilitation in clinical practice where it is crucial to assure a motor recovery that translates to a broad spectrum of motor activities. The translational value of our study is also enhanced by the results of the combined treatment on the skilled reaching test. In humans, the kinematic analysis of skilled reaching is used in clinical practice to provide objective and quantitative information about forelimb motor function after injury. Despite different behavioral specializations, skilled reaching in humans shares many components with rodents (*Klein et al., 2012*). Indeed, *ArcLen*, *Mean Speed* and *Smoothness* parameters extracted from our analysis show important similarities with observed trajectories in stroke patients (*Cirstea and Levin, 2000*; *Panarese et al., 2012*).

Overall, in this manuscript we have defined a rehabilitative protocol that is highly effective in obtaining a complete 'true recovery' (*Zeiler and Krakauer, 2013*; *Reinkensmeyer et al., 2016*) of motor function. Importantly, we showed treatment efficacy in kinematic parameters of reaching that may generalize to patient populations. The results point to a synergistic effect of contralesional activity downregulation and robotic training after small ischemic infarcts in motor cortex.

# Materials and methods

**Key resources table**

| Reagent type (species) or resource | Designation | Source or reference | Identifiers | Additional information |
|---|---|---|---|---|
| Genetic reagent (*Mus musculus*) | 21B6.Cg-Tg (Thy1-ChR2/EYFP)18Cfng/J | Jackson Laboratories | IMSR_JAX:007612 | |
| Antibody | NeuN | Millipore | RRID:AB_11205592 | MILLIPORE:ABN90 Guinea Pig polyclonal; diluition (1:1000) |
| Antibody | Parvalbumin | SynapticSystems | RRID:AB_2156476 | SYSY:195004 Guinea Pig polyclonal; diluition (1:300) |
| Antibody | Somatostatin | Millipore | ID_MILLIPORE:MAB354; clone YC7 | Rat monoclonal; diluition (1:400) |
| Antibody | Myelin Basic Protein | Millipore | RRID:AB_2255365 | MILLIPORE:AB980 Rabbit polyclonal; diluition (1:500) |
| Antibody | intact and BoNT/E-truncated SNAP-25 | Other | | Ref: https://doi.org/10.1016/j.neuroscience.2010.04.059; https://doi.org/10.1523/JNEUROSCI.0772-07.2007 |

*Continued on next page*

*Continued*

| Reagent type (species) or resource | Designation | Source or reference | Identifiers | Additional information |
|---|---|---|---|---|
| Antibody | Hoechst 33258 | Thermo Fischer | RRID:AB_2651133 | Thermo Fisher Scientific Cat# H3569 |
| Chemical compound, drug | Rose Bengal | Sigma-Aldrich | ID_ALDRICH:330000 | |
| Chemical compound, drug | 6-cyano-7-nitroquinoxaline-2,3-dione (CNQX) | Tocris | ID_PubChem:3721046 | |
| Chemical compound, drug | CGP 55845 | Tocris | ID_PubChem:5311042 | |
| Chemical compound, drug | Botulinum Neurotoxin E (BoNT/E) | Other | | Kindly provided by Thomas Binz (Hannover, Germany); ref: https://doi.org/10.1523/JNEUROSCI.4402-04.2005; https://doi.org/10.1523/JNEUROSCI.0772-07.2007 |
| Commercial Assay or kit | PlexBrightOptogenetic Stimulation System | PlexonInc | | PlexBright LD-1 Single Channel LED Driver with 456 nm Table-top LED Module |
| Commercial Assay or kit | OmniPlex D Neural Data Acquisition System | PlexonInc | | |
| Software, algorithm | LabWindows/CVI | http://www.ni.com/lwcvi/i/ | | |
| Software, algorithm | MATLAB | http://www.mathworks.com/products/matlab | RRID:SCR_001622 | |
| Software, algorithm | Offline Sorter | PlexonInc | RRID:SCR_000012 | |
| Software, algorithm | NeuroExplorer | PlexonInc | RRID:SCR_001818 | |
| Software, algorithm | Stereo Investigator | MBF Bioscience | RRID:SCR_002526 | |
| Software, algorithm | ImageJ | NIH | RRID:SCR_003070 | |
| Software, algorithm | R | https://www.r-project.org/ | RRID:SCR_001905 | |
| Software, algorithm | G Power Software | http://www.gpower.hhu.de/ | RRID:SCR_013726 | version 3.1.5 |
| Software, algorithm | Kinematic Analysis | other | | Ref: https://doi.org/10.1177/1545968314545174 |
| Other | Optic Fiber 200 µm Core 0.39 NA | ThorlabsInc | ID_THORLABS:M83L01 | |
| Other | 16 channels linear probes | NeuroNexus | | |
| Other | USB DAQ board | National Instruments | ID_NI:USB-6212 BNC | |
| Other | M-Platform | other | | Ref: https://doi.org/10.1177/1545968313506520 |

## Experimental design

All procedures were performed in compliance with the EU Council Directive 2010/63/EU on the protection of animals used for scientific purposes, and approved by the Italian Ministry of Health, protocol number DGSAF0015924-16/06/2015. A total of 63 C57BL6J mice were used (22–27 g, age 8–10 weeks). We estimated the minimum number of animals required to measure an improvement in the motor outcomes, based on the results of our previous paper (*Alia et al., 2016*). From data represented in *Figure 7* of *Alia et al. (2016)*, we estimated an effect size of 1.75 that was used to calculate the minimum number of animals necessary for this study to obtain a power >80%. We calculated that a number of 5 animals per group was enough to have a power of ≈ 81% so we considered a minimum of 5 animals per treatment group. Because of the high number of groups in the study, we organized different experimental sessions with different cohorts of mice. In each cohort we tested the homogeneity of the baseline performance and randomly assigned the animals to the different experimental groups. The randomization was performed through a computerized random numbers procedure and conducted independently of the study investigators. To ensure internal

control, we have included at least two stroke untreated animals to be used as controls in each cohort. Power calculations were performed with G Power Software (version 3.1.5).

In order to investigate interhemispheric connectivity after stroke, B6.Cg-Tg (Thy1-ChR2/EYFP) 18Cfng/J (Jackson Laboratories, USA) were used for electrophysiological recordings following optogenetic stimulation in the opposite hemisphere (Sham group n = 7, Stroke 30 days n = 9, Stroke 5 days n = 5, Robot+BoNT n = 5). For pharmacological assessment of CGP55845 effect on wild type mice, additional n = 7 Sham and n = 7 Stroke untreated Thy1-ChR2 mice were recorded, for a total number of 40 B6.Cg-Tg (Thy1-ChR2/EYFP)18Cfng/J mice.

To assess the impact of rehabilitative treatments on post-stroke recovery, a pool of 42 C57 BL6/J mice was tested in the Gridwalk and Schallert cylinder tests to establish a pre-injury baseline performance. All of these animals underwent a surgical procedure but n = 5 mice received a sham surgery while n = 37 received a stroke in CFA and then were randomly assigned to the four different groups: untreated stroke group (n = 11), Robot group (n = 10), BoNT group (n = 5), Robot+BoNT group (n = 11) and tested at day 2, 9, 16, 23 and 30 post-lesion in Gridwalk and Schallert Cylinder test. For the follow-up experiment, n = 5 Robot and n = 5 Robot+BoNT animals were kept untreated for 10 days after day 30 behavioral assessment and tested again in Gridwalk and Schallert Cylinder test at day 40 post-lesion.

Before undergoing the ischemic lesion, a subgroup of these animals were trained for two weeks in a custom-made skilled reaching apparatus (*Lai et al., 2015*) until the performance showed a plateau, to collect at least three baseline sessions. Mice that reached criterion performance were divided in two groups (n = 6 stroke untreated and n = 5 Robot+BoNT). These animals were tested after lesion or sham treatment in the reaching apparatus for end point and kinematic analysis at the same time points (see below for details about behavioural tests).

At the end of the experiment (30 days post surgery), all animals were sacrificed for immunohistochemical analyses of plasticity markers with additional n = 2 Sham animals which did not perform behavioural tests.

Finally, to test the effectiveness of Botulinum Neurotoxin E (BoNT/E) injection, a total number of 19 mice were used for Western blotting (n = 9), immunohistochemical analysis (n = 5) and behavioural tests (n = 5).

All the behavioural and electrophysiological analyses were conducted as blinded, randomized experiments. For anatomical investigations, countings were blind to spontaneous and rehabilitated groups but the evident presence of the lesion prevented to be blind to stroke and sham groups.

The entire study included a total number of 103 (40 B6.Cg-Tg (Thy1-ChR2/EYFP)18Cfng/J and 63 C57BL6/J) mice.

## Photothrombotic lesion

The photothrombotic lesion was induced as previously described (*Lai et al., 2015*). Briefly, animals were anesthetized with Avertin (20 ml/kg, 2,2,2 tribromoethanol 1.25%; Sigma-Aldrich, USA) and placed in a stereotaxic apparatus. After a midline scalp incision, the bone was carefully dried and cleaned. Rose Bengal (0.2 ml of a 10 mg/ml solution in PBS; Sigma Aldrich) was injected intraperitoneally. After 5 min, the brain was illuminated through the intact skull for 15 min using a cold light source (ZEISS CL 6000, Germany) linked to a 20X objective that was positioned 0.5 mm anterior and 1.75 mm lateral from Bregma, that is, in correspondence with the caudal forelimb area (*Tennant et al., 2011*). Sham animals underwent scalp incision and Rose Bengal injection but no light irradiation. For consistency, the cortical lesion was always induced in the right hemisphere and the same hemisphere was used in sham animals as control.At the end of the surgery, the skin was sutured and mice were allowed to awaken from anaesthesia.

## Electrophysiological recordings after contralateral light stimulation

For electrophysiological analyses of interhemispheric coupling, we used Thy1-ChR2 Transgenic mice (B6.Cg-Tg (Thy1-ChR2/EYFP)18Cfng/J, Jackson Laboratories, USA) that express the gene encoding for ChR2 under the Thymus cell antigen-1 (Thy-1) promoter (*Hira et al., 2013*). Sham and stroke mice at 30 days were anesthetized with an initial cocktail of ketamine (100 mg/kg, i.p.) and xylazine (10 mg/kg, i.p.) that was supplemented with additional doses to maintain the plane of anesthesia.

Each animal was then placed in a stereotaxic apparatus and a midline incision was made to expose the skull and the sutures.

A 3 × 3 mm craniotomy centered at 2 mm anterior and 1.2 mm lateral to Bregma (corresponding to the RFA, *Tennant et al., 2011*; *Vallone et al., 2016*) was performed in both hemispheres of the anesthetized animal. The dura mater was left intact and a recording chamber of dental cement (Ivoclar Vivadent Inc., USA) was made to preserve and moisten the tissue with saline or for local drugs application. The ground electrode was positioned in an additional small craniotomy over the cerebellum. The tip of the optic fiber was positioned stereotactically over the dura mater of the RFA in one hemisphere and just leaned over the tissue.

Optogenetic stimulation was delivered by means of PlexBright Optogenetic Stimulation System (PlexonInc, USA) with a PlexBright LD-1 Single Channel LED Driver (PlexonInc, USA) and a 456 nm Table-top LED Module connected to a 200 $\mu$m Core 0.39 NA optic fiber (ThorlabsInc, USA). The maximum emission power of the fiber optic was assessed before each experiment with PlexBright Light Measurement Kit; for each experiment the maximum emission power was about 10 mW (79.55 mW/mm$^2$). A safe range for in vivo experiments is about 75 mW/mm$^2$ for short pulses (0.5–50 ms) (*Cardin et al., 2010*), so we never delivered optogenetic stimuli over 31.82 mW/mm$^2$ (40% of maximum emission power). Stimulation parameters were controlled by a custom-made software developed in LabWindows/CVI (National Instruments, USA) through a USB DAQ board (NI USB-6212 BNC, National Instruments, USA).

Neuronal activity was recorded in the other hemisphere by means of 16 channels linear probes (NeuroNexus, USA), connected to the OmniPlex D Neural Data Acquisition System (PlexonInc, USA). Wide Band (WB) signals were acquired at 40,000 Hz, amplified 1K, and band-pass filtered (0.03–12,000 Hz). Local Field Potentials (LFP) and continuous spike signals (SPKC) were computed online by band-pass filtering the WB signals (0.03–300 Hz and 300–10,000 Hz, respectively) and referred to the ground electrode in the cerebellum. Multi-unit activity (MUA) was computed offline by further processing the SPKC signals, as in *Stark and Abeles (2007)*. Briefly, MUA was estimated by computing the sample-by-sample RMS. LFPs from the 16 channels were used to compute and plot inverted current source density (CSD) with step method, as in *Pettersen et al. (2006)* using the MATLAB toolbox CSDplotter. With this configuration, we first recorded neuronal activity at increasing stimulation powers, from values that were under threshold until reaching a plateau of neural response. Stimulations were repeated 15 times for each power intensity, spaced by 5 s. For all of the animals in each group, the quantification of the evoked FP and MUA was performed following stimulation with single light pulses at 3 mW (23.865 mW/mm$^2$, 30% of maximum emission intensity) power. For the double-pulse protocol, a common tool to investigate short-term plasticity (*David-Jürgens and Dinse, 2010*), we set on the stimulation power that corresponded approximately to the 75% of the neuronal response in terms of amplitude of the FP. The double-pulse protocol consisted of 2 light pulses delivered with the same power amplitude, lasting 1 ms each and spaced by the following Inter Stimulus Intervals (ISI): 50, 100, 200 ms. Stimulations were repeated 15 times for each ISI, spaced by 5 s. In order to verify the post-synaptic nature of the recorded FP, the AMPA receptor antagonist 6-cyano-7-nitroquinoxaline-2,3-dione (CNQX, 1 mM; Tocris, UK) was topically applied over the craniotomy, without removing the electrode. Likewise, to identify the receptor subtypes involved in the interhemispheric inhibition in stroke animals, the selective GABA-B antagonist CGP 55845 (10μM, Tocris, UK) was applied. Neural signals were acquired at regular time intervals up to 30 min to verify the effect and the penetration of the drug in the cortical layers.

At the end of the experiment the animal was sacrificed and, eventually, its brain was taken for histological analyses.

Data were analyzed offline with Offline Sorter and NeuroExplorer software (PlexonInc, USA) and with custom made Matlab User Interfaces (Matlab, Matworks). Specifically, for the analysis of the single-pulse FP, we quantify separately the amplitude of the negative- and positive-going components by measuring the baseline-to-peak value. For the PP analysis, based on previous reports in literature (*Schmidt et al., 2012*) we analyzed the peak-trough amplitude, i.e. to avoid any interference of the altered baseline preceding the second stimulation.

## Silencing of the healthy hemisphere with Botulinum Neurotoxin E

Toxin injections were performed in the same surgical session of photothrombotic lesions; after 15 min of illumination that caused the Rose Bengal activation, the dura mater was exposed over the

CFA in the non-injured hemisphere by means of a dental drill. We injected 500 nl of BoNT/E (80 nM) or vehicle divided in two separate injections of 250 nl at (i) + 0.5 anteroposterior, +1.75 mediolateral and (ii) + 0.4 anteroposterior, +1.75 mediolateral by means of heat pulled glass micropipettes (Harvard Apparatus, Holliston, MA) at 700 µm cortical depth. After toxin infusion, the micropipette was left in place for at least 5 min. After surgery animals were sutured and treated with paracetamol (100 mg/kg) in drinking water for four post-operation days.

## Robotic rehabilitation

Mice were trained by means of a robotic platform, the *M-Platform*, as we showed in our previous work (*Spalletti et al., 2014*). Briefly, the robotic device comprises a linear actuator, a 6-axis load cell, a precision linear slide with a controlled friction system and a custom-designed handle that was fastened to the left wrist of the animal. One end of the handle was screwed on the load-cell for loss-less transfer of the forces to the sensor, whereas the other end formed a support for the animal wrist. The animal was kept in a U-shaped restrainer, and its head was stabilized by means of a cemented post. The daily training consisted in the execution of 10–15 sessions of forelimb retraction performed by the animals. First, the linear motor pushed the handle and extended the mouse forelimb by 10 mm (full upper extremity extension). Then, the motor decoupled from the slide and the mouse could initiate the task. If able to overcome a force threshold of 0.2 N, i.e. static friction, the animal voluntarily pulled the handle back (i.e. forelimb flexion back to the starting position). Upon successful completion of the task, the animal was given access to a liquid reward, i.e. 10 µl of sweetened condensed milk, before starting a new session. All of the experimental sessions were recorded by a video camera placed parallel to the coronal plane of the mouse. Position and speed signals were subsequently extracted from the video recordings and synchronized with the force signals recorded by the load-cell. From these kinematic and kinetic signals, a series of parameters were automatically computed to describe detailed motor performance on the platform including the t-target (i.e., the time spent by the animal to accomplish a single retraction task) and the number of attempts to move the handle, with 'attempts' defining the force exerted by the mouse not sufficient to overcome the static friction. The computation and statistical analysis of these parameters were performed using custom-made algorithms developed in Matlab (Mathwork, USA) (*Spalletti et al., 2014*).

Before the lesion mice were allowed let free to explore the platform and gradually habituated to be restrained and to receive the reward. In the same surgery of the photothrombotic lesion, a metal post (length 8 mm, diameter 2 mm, weight 0.2 g) was placed on the occipital bone and fixed by means of dentistry cement (Super Bond C and B, Sun Medical Company, Japan). Since the surgery was minimally invasive, mice recovered in 24 hr. Thus, on the following day, mice were already head restrained on the platform, with their wrist positioned in the handle while regular rewards were provided. Thanks to this habituation phase, mice did not show stress or fear behaviour and easily get used to head restraining.

The day after, that is, 2 days post lesion, they were tested again in behavioural tests and kinematic values were acquired. After these tests, mice were restrained again on the platform and only few sessions of retraction task were performed. The animals started the daily rehabilitative treatment at day five post lesion and continued it until day 30 (4 weeks), performing 10–15 forelimb retraction sessions for 4 days a week. For the follow-up experiment, after behavioral assessment at day 30, the robotic therapy was interrupted until day 40 when animals were tested again in the Gridwalk and Schaller Cylinder tests.

## Motor tests

Motor performance of all the experimental groups was assessed in baseline condition and then once a week at days 2, 9, 16, 23 and 30 post-lesion (with the additional day 40 time point for the follow-up experiments) using two classical behavioral tests, Gridwalk and Schallert Cylinder test. A sub-group of animals (see above) were also tested in the Skilled Reaching Test, followed by kinematic analysis of the paw trajectories.

*Gridwalk Test:* animals were allowed to walk freely for 5 min on an elevated grid (32 × 20 cm, with 11 × 11 mm-large openings) and the task was video-recorded. The video recordings were analyzed off-line by means of a custom-designed Graphical User Interface implemented in Matlab

(*Lai et al., 2015*), to assess correct steps and foot-faults, that is, steps not providing body support, with the foot falling into grid hole, blind to the experimental group. The percentage of foot faults for each limb was then calculated, as previously described (*Lai et al., 2015*).

*Schallert Cylinder Test:* animals were placed in a Plexiglas cylinder (8 cm diameter, 15 cm height) and recorded for five minutes by a video-camera placed below the cylinder. Videos were analyzed frame by frame and the spontaneous use of both forelimbs was assessed during exploration of the walls, by counting the number of contacts performed by the paws of the animal. The experimenter was blind to the experimental group. For each wall exploration, the last paw that left and the first paw that contacted the wall or the ground were assessed. In order to quantify forelimb-use asymmetry displayed by the animal, an Asymmetry Index was computed, according to *Lai et al. (2015)*.

*Skilled Reaching Test and Kinematic Analysis:* The percentage of correct movements and the kinematic analysis of the whole reaching movements were performed as previously described (*Lai et al., 2015*). Briefly, animals (food deprived for 15 hr) were placed in a testing chamber with plastic walls and trained to perform a skilled reaching task with their preferred paw, which had to pass through a small frontal rectangular aperture (0.5 × 1.3 cm) to grasp and retrieve food pellets. The task was recorded by a high frame-rate video camera (120 frames per second, Hero 3, GO-Pro, USA), placed on the side of the testing chamber thus allowing for a sagittal view of the animal.

The number of correct (i.e., a reach and grasp movements ending with pellet eating) and incorrect movements (i.e., when the mouse passed by the frontal window and reached the pellet but either missed it or dropped the pellet after grasping it) were manually assessed. Then the percentage of incorrect grasping was calculated on the total attempts, defined as every time that the paw crossed the frontal window.

Off-line reconstruction of paw trajectories was performed by a semi-automated algorithm based on colour contrast analysis, as described in *Lai et al. (2015)*. Briefly, the algorithm tracked the trajectories of the preferred paw on the sagittal (*x, y*) plane by identifying the position of the paw previously painted with a green non-toxic dye (Stabilo Boss, Stabilo, Germany). To ensure consistency, only trajectories from successful trials (i.e. correct movements) were considered. Changes in the trajectories were quantified by the length of the whole trajectory (ArcLen), by the area enclosed by the reaching and retracting movement (Area Under the Curve, AUC), by the average value of the tangential velocity profile (Mean Speed) and by the number of peaks in the tangential velocity profile (Smoothness). A detailed description of kinematic parameters is provided in *Lai et al. (2015)*.

## Immunohistochemical analysis

For immunohistochemical analysis of plasticity markers, animals were transcardially perfused with 4% paraformaldehyde. Brains were cut using a sliding microtome (Leica, Germany) to obtain 50 μm thick coronal sections that were used for immunostaining of NeuN (1:1,000, Millipore, Germany), Parvalbumin (1:300, Synaptic Systems, Germany), Somatostatin (1:400, Millipore, Germany) and Myelin Basic Protein (1:500, Millipore, Germany). The number of Parvalbumin- and Somatostatin-positive neurons was analyzed using a fluorescence microscope (Zeiss, Germany) with a 10x objective, counting in a 200 μm wide cortical column drawn at the medial and lateral edge of the ischemic tissue by Stereo Investigator software (MBF Bioscience, USA). Three sections per animal were analyzed. For MBP analysis, images were acquired using a 10x objective and analyzed offline drawing 200 μm wide columns at the lateral edge of the ischemic lesion using Image J software (National Institutes of Health, USA) and measuring mean fluorescence for each column. To evaluate the effect of BoNT/E injected into the motor cortex, we stained cortical sections with antibodies recognizing either the intact or BoNT/E-truncated forms of SNAP-25 (*Caleo et al., 2007*; *Antonucci et al., 2010*). The analysis was performed 2 days after BoNT/E injections and three sections per animal were analyzed. To quantify the lesion volume, 1 out of every six sections was stained with Hoechst 33258 (Sigma-Aldrich, USA). The ischemic region was contoured using Stereo Investigator software (MBF Bioscience, USA) with a 10x objective and its area measured. The lesion volume for each animal was calculated by summing up all damaged areas and multiplying the number by section thickness and by 6 (the spacing factor). A total infarction volume in mm$^3$ is given as the mean ± standard error of all analyzed animals (n = 4 stroke untreated and n = 4 Robot +BoNT).

## Statistical analysis

All statistical tests were performed using SigmaPlot 11.0 (Systat Software Inc, USA) and the free software statistical environment 'R' (*R Development Core Team, R. F. F. S. C, 2008*).

For behavioural tests (Gridwalk test, Schallert Cylinder test and skilled reaching test) One-Way and Two Way Repeated Measures ANOVA were used, followed by a Tukey test. For comparison between groups, Two-Way ANOVA was used, followed by a Tukey test. For electrophysiological data and kinematics parameters, Two Way ANOVA was used followed by Tukey test, while paired T-test was used to evaluate FPs before and after CGP application. For immunohistochemical analysis T test was used. All statistical analyses were performed on raw data (alpha value 0.05). No samples were excluded from analysis.

## Acknowledgements

We thank Thomas Binz (Hannover, Germany) for the kind gift of BoNT/E. Marta Pietrasanta contributed to data collection for *Figure 4*. We also thank Francesca Biondi for the excellent animal care and Renzo Di Renzo for technical support.

## Additional information

### Funding

| Funder | Grant reference number | Author |
| --- | --- | --- |
| Fondazione Pisa | 158/2011 | Matteo Caleo |
| Regione Toscana | | Silvestro Micera<br>Matteo Caleo |
| European Commission | Horizon 2020 Research and Innovation Program 720270 (HBP SGA1) | Silvestro Micera<br>Matteo Caleo |
| ERC Advanced Grant 2015 | 692943 | Matteo Caleo |

The funders had no role in study design, data collection and interpretation, or the decision to submit the work for publication.

### Author contributions

Cristina Spalletti, Conceptualization, Data curation, Formal analysis, Investigation, Writing—original draft, Writing—review and editing; Claudia Alia, Conceptualization, Data curation, Formal analysis, Investigation, Writing—review and editing; Stefano Lai, Alessandro Panarese, Data curation, Software, Formal analysis, Writing—review and editing; Sara Conti, Investigation; Silvestro Micera, Matteo Caleo, Conceptualization, Funding acquisition, Methodology, Project administration, Writing—review and editing

### Author ORCIDs

Cristina Spalletti http://orcid.org/0000-0003-2007-4026
Silvestro Micera https://orcid.org/0000-0003-4396-8217
Matteo Caleo http://orcid.org/0000-0002-4333-6378

### Ethics

Animal experimentation: All procedures were performed in compliance with the EU Council Directive 2010/63/EU on the protection of animals used for scientific purposes, and approved by the Italian Ministry of Health, protocol number DGSAF0015924-16/06/2015.

### Decision letter and Author response

Decision letter https://doi.org/10.7554/eLife.28662.041
Author response https://doi.org/10.7554/eLife.28662.042

# Additional files

## Supplementary files
• Transparent reporting form
DOI: https://doi.org/10.7554/eLife.28662.037

## Major datasets
The following dataset was generated:

| Author(s) | Year | Dataset title | Dataset URL | Database, license, and accessibility information |
|---|---|---|---|---|
| Lai S, Spalletti C | 2017 | Data Repository - Combining robotic training and inactivation of the healthy hemisphere restores pre-stroke motor patterns in mice | https://dx.doi.org/10.6084/m9.figshare.5028755 | Available at figshare under a CC0 Public Domain licence |

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
