## [Decision Letter]

Thank you for submitting your article "Combining robotic training and inactivation of the healthy hemisphere restores pre-stroke motor patterns in mice" for consideration by *eLife*. Your article has been reviewed by three peer reviewers, one of whom is a member of our Board of Reviewing Editors and the evaluation has been overseen by Richard Ivry as the Senior Editor. The following individuals involved in review of your submission have agreed to reveal their identity: Heidi Johansen-Berg (Reviewer #1).

The reviewers have discussed the reviews with one another and the Reviewing Editor has drafted this decision to help you prepare a revised submission.

Summary:

The authors tested the effects of combining robotic training and inactivation of the contralesional healthy cortex on recovery after experimental stroke in mice. They hypothesized that this strategy would release transcallosal inhibition of the perilesional sensory/motor cortex, resulting in enhanced motor recovery after stroke. The paper presents an impressive series of experiments, involving behavioral testing, electrophysiological recordings and immunohistochemical analyses, in which different factors were systematically investigated. Results demonstrate that inactivation of the contralesional cortex leads to disinhibition of the ipsilesional cortex, which in combination with rehabilitation treatment, significantly improved recovery of motor functions. These findings support the reasoning behind clinical studies that apply inhibitory TMS or tDCS to the unaffected hemisphere, combined with rehabilitation treatment, to promote recovery in stroke patients.

Despite the large amount of acquired data, some important aspects were not detailed or discussed. In order to judge whether the conclusions of the paper hold, the reviewers require responses to fairly extensive revisions, detailed below. Responses to some of these issues, relating to study design, lesion type, and statistical analyses, will determine whether or not we wish to proceed with the paper at *eLife*.

Essential revisions:

1) The study relies on the concept of an interhemispheric imbalance after stroke, but this concept may not always apply and conflicting results have been reported (see reviews by Di Pino et al., (2014) and Boddington and Reynolds, (2017)). The rationale to target transcallosal pathways therefore requires more explanation. Would this be more promising or effective in comparison to facilitatory ipsilesional stimulation strategies?

2) Sample sizes were 'determined by previous behavioral experiments and pilot experiments'. This needs to be clarified in more detail. Also, the authors report that animals were randomly assigned to experimental groups. How was randomization executed and is this the reason that group sizes are not equal (e.g. spontaneous group: n=11; robot group: n=5; BoNT group: n=5; Robot+BoNT group: n=6 (subsection “Experimental design”))? Sample sizes in a treatment study should be based on an appropriate power calculation. Absence of a priori power calculation would be a serious shortcoming of the paper.

3) Disturbed transcallosal signal transmission may also be explained by direct injury to the corpus callosum. Did the photothrombosis induce any damage to the underlying white matter (see Lee et al., (2007), who reported partial white matter damage in the same model)? The authors should report the spatial extent of the lesion. I encourage the authors to show a lesion incidence map and/or examples of the lesion on histological sections.

4) The effect of any type of treatment is strongly dependent on the time after stroke. The authors should provide arguments for their chosen treatment onset time, which was 5 days after stroke for robotic training, while contralesional inactivation was induced immediately after stroke. Do they have any data (or at least views) on effects of earlier robotic training or later contralesional inactivation and/or robotic training? This is relevant because an interhemispheric imbalance after stroke may not be a static phenomenon.

5) It is important to know if positive effects of rehabilitative treatments last beyond the treatment period. In the present study, serial performance assessments were done one day after each treatment series, up to 30 days after stroke. Do the authors have any evidence of enduring effects after this period?

6) Throughout the paper statistical tests appear to have been only made within group. So for a given treatment, for example, measures at post stroke/treatment timepoints are compared to baseline measures within group. However, no direct comparisons appear to have been made between groups. In order to assess whether, e.g. robot treatment plus BoNT/E is better than either treatment alone, it would be necessary to directly compare the groups, rather than simply reporting that more timepoints are significantly different from baseline within the combined group (e.g. this is relevant for Figure 4, Figure 6, Figure 7, Figure 8). Similarly, if wanting to infer that a function is 'rescued' (e.g. Figure 10) then the authors should statistically compare the treated and untreated groups, rather than simply reporting lack of significant difference between sham and treated groups. Despite the lack of direct statistical comparison between groups, much of the discussion tried to make inferences about, for example, the special consequences of the combined therapy. These conclusions could be much stronger if the combined therapy were compared directly to the separate therapies.

7) It is unclear how to interpret the evoked potential. There appears to be two phases. The CNQX blocks all of it. However, GABAB blockers abolish the secondary phase without affecting the first. Doesn't that imply that the second phase is driven by local excitation in the RFA and not per se due to "transcallosal inhibition"?

8) It seems worth modelling the two phases to show what is GABAB and what is the initial volley of excitation. For example, what is the typical residual with such GABAB blockade? I would suggest a detailed description of the evoked potential and effects of cnqx/GABAB blockers effects early in the manuscript.

9) In stroke, it seems that both the first and the second phase of the evoked potential is modified. How do the authors interpret this? Is there also weaker excitation? Did they look at GABAB blockers in the healthy condition?

---

## [Author Response]

Essential revisions:1) The study relies on the concept of an interhemispheric imbalance after stroke, but this concept may not always apply and conflicting results have been reported (see reviews by Di Pino et al., (2014) and Boddington and Reynolds, (2017)). The rationale to target transcallosal pathways therefore requires more explanation. Would this be more promising or effective in comparison to facilitatory ipsilesional stimulation strategies?

We have better introduced the concept of interhemispheric imbalance and the rationale to target transcallosal pathways vs. stimulating the ipsilesional side (Introduction). Plasticity in the lesioned hemisphere plays a major role in post-stroke motor recovery and is a primary target for rehabilitation therapy. Indeed, stimulation of the ipsilesional motor cortex, especially when paired with motor training, facilitates plasticity and functional restoration (Allman et al., 2016; Dodd et al., 2017). On the other hand, the role of the contralesional hemisphere remains highly controversial (Dancause et al., 2015; Talelli et al., 2012; Buetefisch, 2015). Attempts to promote stroke recovery by inhibiting the contralesional hemisphere are based on the interhemispheric competition model, which posits an enhanced transcallosal inhibition exerted by the healthy side over the spared perilesional tissue (Silasi and Murphy, 2014; Boddington and Reynolds, 2017; Plow et al., 2016). However, direct electrophysiological measures of the evolution of interhemispheric inhibition post-stroke are not yet available. Second, inactivation of the healthy hemisphere via either low-frequency, repetitive transcranial magnetic stimulation (rTMS) or cathodal (inhibitory) transcranial direct current stimulation (tDCS) has yielded some positive yet variable effects in clinical trials (Kim et al., 2010; Lefaucheur et al., 2014; Di Pino et al., 2014, Plow et al., 2016). This variability in outcome may depend on the extent of damage (Di Pino et al.; 2014). Indeed, interhemispheric competition appears to dominate in patients with limited damage in the affected hemisphere, while after large lesions the contralesional side vicariates the lost functions (Bradnam et al., 2012; Di Pino et al., 2014). Based on these premises, here we demonstrated directly an enhanced interhemispheric inhibition after focal ischemic lesions, and showed that suppressing contralesional activity combined with robotic training improves forelimb motor function.

2) Sample sizes were 'determined by previous behavioral experiments and pilot experiments'. This needs to be clarified in more detail. Also, the authors report that animals were randomly assigned to experimental groups. How was randomization executed and is this the reason that group sizes are not equal (e.g. spontaneous group: n=11; robot group: n=5; BoNT group: n=5; Robot+BoNT group: n=6 (subsection “Experimental design”))? Sample sizes in a treatment study should be based on an appropriate power calculation. Absence of a priori power calculation would be a serious shortcoming of the paper.

We thank the reviewers and we now report a more detailed description of our power analysis. Since the primary outcome of the study was the efficacy of the treatment in the motor outcomes, we estimated the minimum number of animals based on the results of our previous paper (Alia et al., 2016), where we demonstrated that a systemic treatment acting on the inhibitory system was able to ameliorate performance in the Gridwalk test. Here, we supposed our treatment could lead to similar or even better performance in the motor outcome. From data represented in Figure 7, panel A (Alia et al., 2016), we estimated an effect size of 1.75 that was used to calculate the minimum number of animals necessary for this study to obtain a power > 80%. We calculated that a number of 5 animals per group was enough to have a power of ≈81% so we considered a minimum of 5 animals per treatment group. Because of the high number of groups in the study, we organized different experimental sessions with different cohorts of mice. In each cohort we tested the homogeneity of the baseline performance and randomly assigned the animals to the different experimental groups. The randomization was performed through a computerized random numbers procedure and conducted independently of the study investigators. To ensure internal control, we have included at least two stroke untreated animals to be used as controls in each cohort, and this is why the sample size of the spontaneous group is higher. To respond to the request for a follow-up effect of the treatment, for this revision we made additional cohorts of animals and now the number of mice for robot and Robot+BoNT groups is increased and so is the power of the analysis. Power calculations were performed with G Power Software (version 3.1.5).

Power analysis and randomization are now better described in Materials and methods section.

3) Disturbed transcallosal signal transmission may also be explained by direct injury to the corpus callosum. Did the photothrombosis induce any damage to the underlying white matter (see Lee et al., (2007), who reported partial white matter damage in the same model)? The authors should report the spatial extent of the lesion. I encourage the authors to show a lesion incidence map and/or examples of the lesion on histological sections.

The photothrombotic lesion typically comprises all cortical layers, with no involvement of the underlying white matter (Figure 1 of the revised manuscript, Nissl staining). In about 25% of the cases, we found limited damage in the dorsal aspect of the corpus callosum (myelin basic protein – MBP – labelling in Figure 1). It is unlikely that the alterations in interhemispheric communication are due to direct callosal damage (Results section), as we found a robust asymmetry between the Field Potential (FP) response recorded in the ipsilesional and contralesional cortex after transcallosal stimulation (Figure 2). Indeed, while FPs recorded in the contralesional side were comparable to those evoked in healthy animals, FPs in the ipsilesional side were robustly affected, pointing to synaptic changes in peri-infarct zone. Based on the indication of the reviewers, we showed examples of the lesion in histological sections (Figure 1).

4) The effect of any type of treatment is strongly dependent on the time after stroke. The authors should provide arguments for their chosen treatment onset time, which was 5 days after stroke for robotic training, while contralesional inactivation was induced immediately after stroke. Do they have any data (or at least views) on effects of earlier robotic training or later contralesional inactivation and/or robotic training? This is relevant because an interhemispheric imbalance after stroke may not be a static phenomenon.

To establish the temporal evolution of interhemispheric functional connectivity after stroke, we recorded FPs evoked by optogenetic stimulation of callosal afferents at an early time point (5 days) following the cortical infarct. We found that the initial, negative component of the evoked FP was dramatically reduced in the perilesional RFA, while there was a trend for enhanced amplitude of positive-going FP (Figure 1). These findings demonstrate an early imbalance in interhemispheric connectivity early after stroke (Results section), and support the choice of a prompt inactivation of the contralesional cortex with BoNT/E, which requires at least 24 hr to become fully active (Antonucci et al., 2008). For robotic training, we chose to start the physiotherapy at day 5, since it is known that early exercise may exacerbate brain injury, via excitotoxicity and generation of reactive oxygen species (Kozlowski et al., 1996; Li et al., 2017). Moreover, it is well established that priming the brain with a “plasticity-inducing” treatment before the beginning of a rehabilitative treatment improves motor performance after brain lesions (Schäbitz et al., 2004; Starkey et al., 2012; Wahl et al., 2014). These issues are now discussed in subsection “Robotic Rehabilitation”.

5) It is important to know if positive effects of rehabilitative treatments last beyond the treatment period. In the present study, serial performance assessments were done one day after each treatment series, up to 30 days after stroke. Do the authors have any evidence of enduring effects after this period?

We agree with the reviewers that the inclusion of a follow-up assessment is important. Accordingly, we have now provided evidence for a persistent effect of the treatment beyond the completion of the rehabilitation protocol. To this aim, two additional cohorts of Robot+BoNT and Robot mice (n = 5 per group) were tested at 40 days, i.e. after a follow-up period of 10 days during which robotic training was suspended. We found that functional gains in the Robot+BoNT group were maintained at 40 days, while no improvements could be detected in the animals undergoing only robotic rehabilitation. The difference between the two groups remained highly significant at 40 days. These new data are shown in Figure 6 and discussed in subsection “Robotic Rehabilitation”.

6) Throughout the paper statistical tests appear to have been only made within group. So for a given treatment, for example, measures at post stroke/treatment timepoints are compared to baseline measures within group. However, no direct comparisons appear to have been made between groups. In order to assess whether, e.g. robot treatment plus BoNT/E is better than either treatment alone, it would be necessary to directly compare the groups, rather than simply reporting that more timepoints are significantly different from baseline within the combined group (e.g. this is relevant for Figure 4, Figure 6, Figure 7, Figure 8). Similarly, if wanting to infer that a function is 'rescued' (e.g. Figure 10) then the authors should statistically compare the treated and untreated groups, rather than simply reporting lack of significant difference between sham and treated groups. Despite the lack of direct statistical comparison between groups, much of the discussion tried to make inferences about, for example, the special consequences of the combined therapy. These conclusions could be much stronger if the combined therapy were compared directly to the separate therapies.

For each motor test, we originally reported the statistics referred to baseline because our purpose was to identify treatments that restore pre-lesion motor performance. We agree that comparisons between the different groups (stroke untreated, BoNT, Robot and Robot+BoNT) are crucial to establish the effectiveness of the combined therapy vs. the separate treatments. Accordingly, we have now shown the comparisons between groups in the behavioural tests and electrophysiological analyses, and we have plotted the motor performances in the Gridwalk and Schallert cylinder tests at the completion of treatment (30 days; Figure 7). In the Gridwalk test, Robot+BoNT group was superior to Robot and untreated stroke but statistically comparable to BoNT/E only. In the Schallert cylinder test, mice receiving the combined therapy displayed markedly improved performances with respect to all the other groups (Figure 7 and Results section). Altogether, these data indicate that the combined treatment yields better results that the separate therapies.

For the data in Figure 10, we have now compared Robot+BoNT vs. stroke untreated and the statistical analysis reveals that several electrophysiological parameters (amplitude of the late positive FP component, paired pulse depression ratio) are indeed rescued by the combined treatment (see Results section).

7) It is unclear how to interpret the evoked potential. There appears to be two phases. The CNQX blocks all of it. However, GABAB blockers abolish the secondary phase without affecting the first. Doesn't that imply that the second phase is driven by local excitation in the RFA and not per se due to "transcallosal inhibition"?

The interpretation of the reviewer is correct. We have used CNQX and GABA-B blockers (Figure 1—figure supplement 1) to demonstrate that the first phase of the evoked potential corresponds to a volley of transcallosal excitation in the RFA, which leads to local activation of GABAergic cells and subsequent GABA-B dependent inhibition of cortical neurons. These data are consistent with neuroanatomical observations indicating that the corpus callosum consists almost entirely of excitatory fibers, which implies that interhemispheric inhibition arises from the activation of local interneurons (such as neurogliaform cells located in layer I; Palmer et al., 2012). These issues are discussed in the Discussion section.

8) It seems worth modelling the two phases to show what is GABAB and what is the initial volley of excitation. For example, what is the typical residual with such GABAB blockade? I would suggest a detailed description of the evoked potential and effects of cnqx/GABAB blockers effects early in the manuscript.

The early negative wave (corresponding to direct transcallosal excitation) is maintained after GABA-B blockade, while the late positive peak in the evoked FP is potently reduced in both healthy and stroke mice (Figure 1—figure supplement 1,C and Figure 3). A description of the waveforms and effects of CNQX/GABA-B blockers is now presented at the beginning of the Results section.

9) In stroke, it seems that both the first and the second phase of the evoked potential is modified. How do the authors interpret this? Is there also weaker excitation?

The electrophysiological data (field potentials and multi-unit activity) indicate weaker direct transcallosal excitation accompanied by an increase in interhemispheric inhibition. The reduction in the negative-going FP and stimulus-induced spiking after stroke may be simply explained by a dampened excitatory drive of pyramidal neurons in the target hemisphere (see Figure 1 and Figure 2). Importantly, callosal fibres also contact inhibitory neurons (such as neurogliaform cells in layer I) which in turn synapse on cortical pyramids. Two non-mutually exclusive possibilities may explain the increased interhemispheric inhibition following stroke: (i) a structural reorganization of callosalfibers after ischemia, with enhanced innervation of layer 1 interneurons, possibly accompanied by a retraction of inputs onto the dendrites of pyramidal cells; (ii) an enhanced synaptic transmission between transcallosally-driven interneurons and cortical pyramids, due to either presynaptic or post-synaptic mechanisms. In particular, previous studies have shown a robust and early upregulation of GABA-B receptors in peri-infarct areas that suggest augmented responsiveness to GABA release. These microcircuit changes are discussed in the Discussion section.

Did they look at GABAB blockers in the healthy condition?

We have now performed electrophysiological recordings on a group of 7 healthy, naive mice before and after application of the GABA-B blocker CPG55845 (Figure 1—figure supplement 1), and the results indicate a significant decrease of the late positive component of the FP with no impact on the early negative wave (see Results section).